# A conserved enhancer regulates *Il9* expression in multiple lineages

Byunghee Koh[1,2], Amina Abdul Qayum[1], Rajneesh Srivastava[3], Yongyao Fu[1,2], Benjamin J. Ulrich[1,2], Sarath Chandra Janga[3] & Mark H. Kaplan[1,2]

Cytokine genes are regulated by multiple regulatory elements that confer tissue-specific and activation-dependent expression. The *cis*-regulatory elements of the gene encoding IL-9, a cytokine that promotes allergy, autoimmune inflammation and tumor immunity, have not been defined. Here we identify an enhancer (CNS-25) upstream of the *Il9* gene that binds most transcription factors (TFs) that promote *Il9* gene expression. Deletion of the enhancer in the mouse germline alters transcription factor binding to the remaining *Il9* regulatory elements, and results in diminished IL-9 production in multiple cell types including Th9 cells, and attenuates IL-9-dependent immune responses. Moreover, deletion of the homologous enhancer (CNS-18) in primary human Th9 cultures results in significant decrease of IL-9 production. Thus, *Il9* CNS-25/*IL9* CNS-18 is a critical and conserved regulatory element for IL-9 production.

[1] Department of Pediatrics and Herman B Wells Center for Pediatric Research, Indiana University School of Medicine, Indianapolis, IN 46202, USA.
[2] Department of Microbiology and Immunology, Indiana University School of Medicine, Indianapolis, IN 46202, USA. [3] Department of Biohealth Informatics, School of Informatics and Computing, Indiana University-Purdue University Indianapolis, Indianapolis, IN 46202, USA. Correspondence and requests for materials should be addressed to M.H.K. (email: mkaplan2@iu.edu)

Gene transcription occurs through a concerted network of transacting factors and *cis*-acting DNA regulatory elements. Apart from gene promoters, directly adjacent to the transcriptional start site (TSS), enhancers exist at more distant sites from the TSS and can be both 5′ and 3′ of the gene locus[1,2]. Enhancers not only facilitate transcription but can confer tissue- and cell-type-specific expression.

In T cells, there are multiple examples of enhancer-specified gene regulation. Among the best characterized are cytokine genes, particularly the *Ifng* and *Il4-Il13* cytokine loci where multiple enhancers and locus control regions have been identified[3–6]. Several of these elements, many based on conserved non-coding sequences (CNSs) or measures of open chromatin such as DNase Hypersensitivity assays, have been deleted in mouse models and have demonstrated function in vitro and in vivo[7–13]. These studies have improved our understanding of how lineage-specific genes are regulated, and how precise gene regulation contributes to immune responses.

**Fig. 1** Identification of a CNS-25 p300-binding element in the *Il9* gene. Naive CD4[+] T cells were isolated from C57BL/6 mice and cultured under the indicated Th cell polarizing conditions. On day 5, cells were harvested for ChIP analysis using p300 antibody. **a** ChIP-seq analysis of p300 in Th9 cells at the *Il9* locus and comparison with ChIP-seq analysis of p300, H3K4me1, and STAT6 in Th2 cells (GSE22104 and GSE40463). Bottom, putative transcription factor-binding motifs in the most conserved regions of CNS-25. **b** Venn diagram indicating the overlap in Th9-enriched genes and p300-binding peaks in Th9 cells. **c** ChIP assay analysis of p300 at the CNS regions of the *Il9* locus in Th cell subsets. Non-conserved sequences −12 and −35 kb were used as negative controls. Percent input depicted are the p300 ChIP values after subtraction of the control IgG ChIP values. **d** p300 and IgG ChIP values at the CNS-25 in Th9 cells. **e** p300 inhibitor added to Th9 cell cultures on day 3. On day 5, cells were harvested and restimulated with anti-CD3 overnight to assess IL-9 production using ELISA. Data are represented as mean ± SEM from three independent experiments (*n* = 3 or 4 per group). **c** One-way ANOVA with a post hoc Tukey test was used to generate *p* values for all multiple comparisons. **p < 0.01, ***p < 0.001. **e** A two-tailed Student's *t* test was used for pairwise comparisons. **p < 0.01

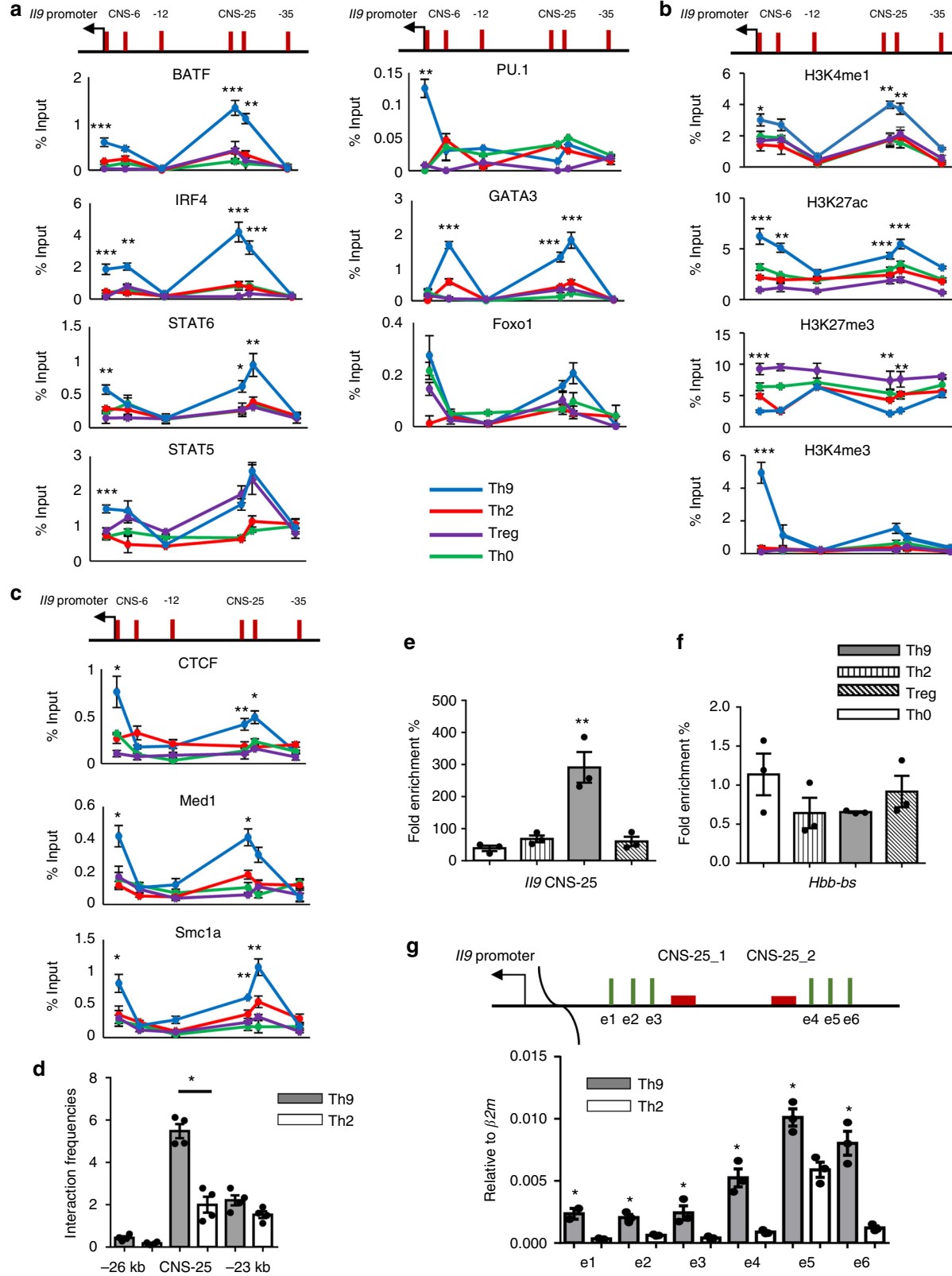

Interleukin (IL)-9 is a pleiotropic cytokine that impacts many cells and is involved in diverse immune responses[14,15]. IL-9, and the cells that produce it, are linked to tumor immunity, immunity to pathogens, allergy, and autoimmune disease. Major producers of IL-9 include T helper (Th) 9 cells and type 2 innate lymphoid cells (ILCs), although mast cells (MCs) and basophils produce

IL-9, and other Th subsets may express limited amounts of IL-9 in some conditions. Despite a growing understanding of the role of IL-9 in immunity and disease, transcriptional regulation of the *Il9* gene is still not well understood. In Th9 cells, the best studied IL-9-secreting cell, expression of IL-9 relies upon a network of transcription factors that include Smad proteins, STAT5, STAT6,

**Fig. 2** *Il9* CNS-25 has enhancer characteristics. Naive CD4[+] T cells were isolated from C57BL/6 mice and cultured under Th cell polarizing conditions. On day 5, cells were harvested for ChIP and 3C analysis (**a–d**) or chromatin accessibility assay (**e**, **f**). **a** The binding of IL-9-inducing transcription factors at the *Il9* locus in Th cell subsets. **b** Histone modifications at the *Il9* locus in Th cell subsets. **c** The binding of DNA loop-mediating proteins at the *Il9* locus in Th cell subsets. **d** DNA loop between CNS-25 and *Il9* promoter in Th9 cells. *Il9* promoter was used as an anchor and −26 kb/−23 kb regions are used as negative controls. **e**, **f** Chromatin accessibility assay fold enrichment was calculated by the ratio of Ct values from qPCR reactions of the nuclease-treated and non-treated samples for the *Il9* locus (**e**) and an irrelevant locus, hemgloblin β chain (*Hbb-bs*) (**f**). **g** The expression of Enhancer RNA near CNS-25 in Th9 and Th2 cells was measured using qRT-PCR with the indicated primer sets. Data are represented as mean ± SEM from three independent experiments (*n* = 4 per group). **a–c**, **e** One-way ANOVA with a post hoc Tukey test was used to generate *p* values for all multiple comparisons. *$p < 0.05$, **$p < 0.01$, ***$p < 0.001$. **d**, **g** A two-tailed Student's *t* test was used for pairwise comparisons. *$p < 0.05$

Forkhead box family members, PU.1, BATF, and interferon regulatory factor 4 (IRF4)[16–27]. These *trans*-factors bind to, and mediate chromatin modifications at, the *Il9* promoter (*Il9p* or CNS1) and at two potential regulatory elements, termed CNS2/CNS + 5.5 and CNS0/CNS-6, that are respectively 5.5 kb downstream and 6 kb upstream of the *Il9* TSS[16,28]. OX40-induced super-enhancers of the *Il9* gene have been identified, and a super-enhancer overlapping CNS + 5.5 was critical for OX40-induced IL-9 production, though it was not clear if this element is important for OX40-indendent IL-9 production[29]. Although most IL-9-promoting transcription factors bind at the *Il9p*[17,30,31], it is still possible that there are additional unidentified regulatory elements of *Il9* expression.

Here we use a combination of bioinformatics and experimental approaches to define an *Il9* enhancer element that regulates IL-9 production within multiple cell lineages. We show that an element 25 kb upstream from the *Il9* gene (termed *Il9* CNS-25) has enhancer characteristics defined by histone modifications, transcription factor binding, and reporter assays. Importantly, mice lacking the CNS-25 element (*Il9*$^{\Delta CNS-25}$ mice) have T cells, MCs, and basophils that produce significantly less IL-9 than wild-type (WT) mice in vitro and in vivo. Deletion of the homologous element in primary human Th9 cells also results in less IL-9 production. Thus, we conclude that *Il9* CNS-25 is a critical conserved enhancer for *Il9* gene expression and will provide an entry point for further understanding the regulation of this immunoregulatory cytokine.

## Results

**Identification of *Il9* CNS-25**. To begin to define additional regulatory elements of the *Il9* gene, we performed chromatin immunoprecipitation and high-throughput sequencing (ChIP-seq) analysis of Th9 cells using Abs to the transcriptional co-activator p300, which is documented to be associated with enhancer elements[32,33]. Analysis identified 2973 p300 occupancy peaks, including within the *Il9* gene, that were predominantly localized to protein-coding genes but spread across gene structure (Fig. 1a and Supplementary Fig. 1a). Of these peaks, 201 genes were identified with differential expression in Th9 cells based on a transcriptome generated from RNA-seq analysis of Th9 gene expression (Fig. 1b). A more detailed analysis of the *Il9* locus identified five occupancy peaks that corresponded to CNS-6 (CNS0), the *Il9p* (CNS1), and CNS + 5.5 (CNS2). Further 5′ of the gene, there were peaks at −20 and −25 kb, with only the −25 kb element showing conservation across species (Fig. 1a). The −25 kb sequence (termed CNS-25) also resides within a region of open chromatin in T cells as cataloged by the ENCODE project (www.encodeproject.org). As an approach to further define the function of these elements, we examined binding of p300 and STAT6, and the presence of H3K4me1 at the locus in Th2 cells using publically available data[33,34]. In Th2 cells, which are permissive for IL-9 production but have limited expression of the gene, we observed peaks at CNS-25, CNS-6, and CNS + 5.5, but not at the promoter, a result consistent with the observed lack of

gene expression (Fig. 1a). To directly compare the amount of p300 binding among Th subsets, we performed p300 ChIP on Th cells differentiated in vitro using polarizing culture conditions. We focused on Th2 and Treg cells that are derived in vitro with IL-4 and transforming growth factor beta (TGFβ), respectively as these cytokines in combination result in Th9 differentiation. As a control, we included Th cells cultured in the absence of any exogenous cytokines (Th0). We observed that p300 binding was most robust in Th9 cells, and at the *Il9p* and CNS-25 (Fig. 1c, d). To simplify subsequent presentation of ChIP data, control Ig values, as indicated in Fig. 1d, have been subtracted from the data presented rather than showing separately. To demonstrate that p300 binding to the locus was functionally important, we differentiated Th9 cells in vitro in the presence or absence of a p300 inhibitor. Culture with the p300 inhibitor decreased IL-9 (Fig. 1e).

**Transcription factor and enhancer activity at *Il9* CNS-25**. The *Il9* CNS-25 spans ~750 bp and contains two smaller regions with more extensive conservation that are 286 and 195 bp in length (Fig. 1a and Supplementary Fig. 1b). Within those segments there are evolutionarily conserved binding sites for STAT proteins, GATA proteins, Smads, Forkhead family members, and an AP-1/IRF consensus element that binds BATF and IRF4 (Fig. 1a and Supplementary Fig. 1b).

To directly test transcription factor binding at the locus in primary T cells, we performed ChIP assays for IRF4, BATF, STAT5, STAT6, PU.1, GATA3, and Foxo1. With the exception of PU.1, which bound predominantly to the promoter region, all of the transcription factors showed binding peaks at CNS-25 in Th9 cells (Fig. 2a). All transcription factors also bound the promoter, although binding of a subset including GATA3 and Foxp1 was not restricted to Th9 cells (Fig. 2a). Binding of other transcription factors was strongest in Th9 cells at the *Il9p* and CNS-25, although STAT5 also bound the CNS-25 in Treg cultures (Fig. 2a). We then examined histone modifications at the locus, observing greater H3K4me1, a modification associated with active enhancers[35], at CNS-25 in Th9 cells (Fig. 2b). In contrast, H3K4me3, a modification associated with gene promoters, was observed at the *Il9p* with only minimal amounts at CNS-25 (Fig. 2b). Reciprocal modifications of H3K27 that are associated with gene activation (acetylation) and repression (tri-methylation) were respectively greatest and least in Th9 cells, with a peak of H3K27Ac at CNS-25 (Fig. 2b). Conversely, Treg cells had the least amounts of H3K27Ac and the greatest H3K27me3 across the locus (Fig. 2b).

A number of proteins are also indicative of higher-order chromatin structures such as looping when bound to regions of DNA. Among these are CTCF, Med 1, a component of the Mediator complex, and Smc1a, a component of the cohesion complex[36–41]. Consistent with the involvement of CNS-25 in regulation of *Il9*, we identified binding of all three transcription factors at CNS-25 and the *Il9p* and only in Th9 cells (Fig. 2c). Using the chromosome conformation capture assay (3C), we further demonstrated DNA looping between CNS-25 and the *Il9*

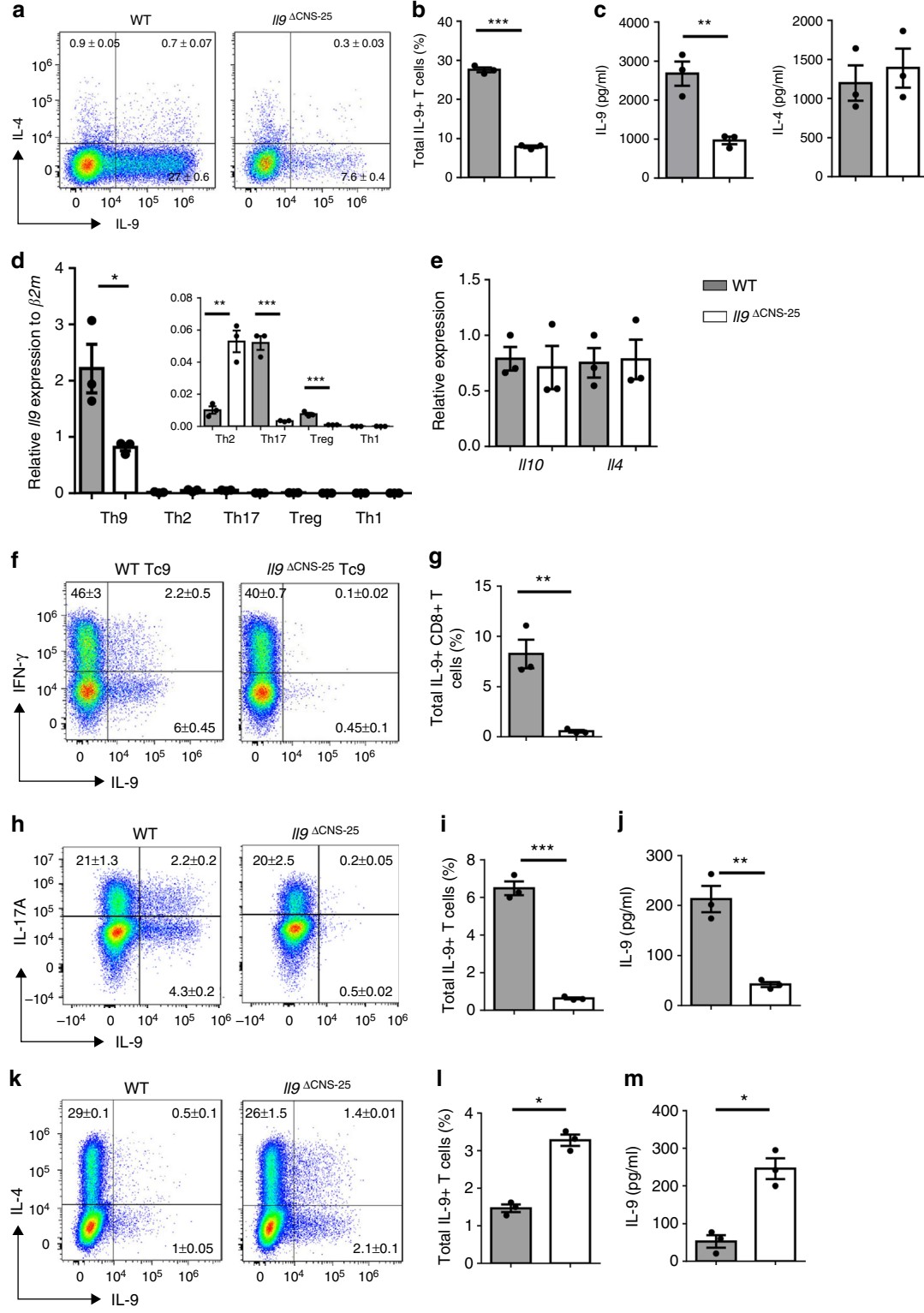

promoter in Th9 cells as being greater than in Th2 cells, and greater than between the promoter and regions adjacent to CNS-25 (Fig. 2d). Transcription factor binding at these sites might also indicate increased chromatin accessibility. To test this directly we performed a nuclease accessibility assay and observed much greater nuclease access in Th9 cells compared to other Th subsets (Fig. 2e, f).

To begin to examine the function of CNS-25, we generated a series of luciferase reporter vector containing CNS-25 in various

positions and orientations, and tested the ability of IRF4 to activate reporter expression in EL4 cells. IRF4 had modest but significant activity on the *Il9p* vector, and a similarly modest ability to induce expression by CNS-25 in either orientation but upstream of the luciferase gene (Supplementary Fig. 2a). When CNS-25 was cloned upstream of *Il9p*, there was a synergistic activation of reporter activity by IRF4, regardless of CNS-25 orientation (Supplementary Fig. 2a). CNS-25 also mediated a modest transactivation when cloned 3′ of the luciferase gene, and

**Fig. 3** CNS-25 deletion impairs IL-9 production in Th9 cells. Naive CD4$^+$ T cells were isolated from wild-type (WT) or $Il9^{\Delta CNS-25}$ mice and cultured different Th cell polarizing conditions. On day 4, cells were harvested for analysis. For intracellular staining, cells were restimulated with PMA/ionomycin for 6 h. Cells were restimulated with anti-CD3 for 6 h or overnight to assess cytokine production using qRT-PCR or ELISA, respectively. **a–c** IL-9 expression in WT or $Il9^{\Delta CNS-25}$ Th9 cultures by flow cytometry (representative dot plots in **a** and average percentage in **b**), or ELISA (**c**). **d** $Il9$ expression of in all Th cell subsets relative to Th9 cells. The inset shows an expanded scale to demonstrate differences in the indicated Th subsets. **e** Cytokine mRNA expression in Th9 cells. **f, g** CD8$^+$ T cells were isolated from WT or $Il9^{\Delta CNS-25}$ mice and cultured under Tc9 cell polarizing conditions. On day 2, cells were harvested and restimulated with PMA/ionomycin for 6 h to assess cytokine production using intracellular staining. Representative dot plots in **f** and average percentages in **g**. **h–j** IL-9 expression in WT or $Il9^{\Delta CNS-25}$ Th17 cultures by flow cytometry (representative dot plots in **h** and average percentage in **i**), or ELISA (**j**). **k–m** IL-9 expression in WT or $Il9^{\Delta CNS-25}$ Th2 cultures by flow cytometry (representative dot plots in **k** and average percentage in **l**), or ELISA (**m**). **a, f, h, k** Numbers indicate the average percentage ± SEM of T cells from 3 mice. All gene expression was normalized to $\beta 2m$ expression. Data are represented as mean ± SEM from three independent experiments ($n = 3$ per group). A two-tailed Student's $t$ test was used for pairwise comparisons. *$p <$ 0.05, **$p < 0.01$, ***$p < 0.001$

further enhanced the ability of the $Il9p$ to activate when cloned on opposite sides of the reporter gene (Supplementary Fig. 2a). CNS-25 significantly increased $Il9p$ reporter activity in both Th0 and Th9 cells, but to a greater degree in Th9 cells (Supplementary Fig. 2b). The data suggest that CNS-25 can enhance $Il9$ promoter activity in primary T cells.

To further examine activity of the enhancer, we examined enhancer RNA (eRNA) transcripts that are commonly found at active transcriptional elements[42–45]. These long non-coding RNAs that are thought to contribute to gene expression are indicative of active regulatory elements. With all six primer sets that amplified sequences both 5′ and 3′ of CNS-25 there was greater transcription observed in Th9 cells than in Th2 cells (Fig. 2g), suggesting that it was more transcriptionally active in IL-9-expressing cells. Together, these data strongly suggest that $Il9$ CNS-25 is an important regulatory element in the $Il9$ gene.

**Germline deletion of the $Il9$ CNS-25 element**. To directly demonstrate that $Il9$ CNS-25 is an important $Il9$ regulatory element, we generated mice that had a deletion of 1.8 kb in the $Il9$ locus including the CNS-25 element (Supplementary Fig. 3a). Homozygous $Il9^{\Delta CNS-25}$ mice developed normally and had normal numbers and percentages of immune cells in spleen and lymph nodes. To determine if the deletion of CNS-25 affected IL-9 production, we differentiated naive CD4$^+$ T cells from WT and $Il9^{\Delta CNS-25}$ mice. We observed IL-9 production as assessed by intracellular staining to be decreased by 75% in the absence of CNS-25 (Fig. 3a, b). Similar decreases were observed using enzyme-linked immunosorbent assay (ELISA) to measure accumulation and quantitative reverse transcription-PCR to measure $Il9$ mRNA (Fig. 3c, d). This effect was specific for $Il9$ as there was no significant difference in expression between WT and $Il9^{\Delta CNS-25}$ Th9 when examining other cytokine production, transcription factors that are required for IL-9 production, or transcription factors that repress $Il9$ expression (Fig. 3c, e, and Supplementary Fig. 3bc). Additionally, the next closest gene to CNS-25, $Fbxl21$, was not detectably expressed in T cells or MCs, and expression in brain tissue was not altered by CNS-25-deficiency (Supplementary Fig. 3d).

Th9 culture conditions can also promote IL-9 production from CD8$^+$ T cells[46,47], and the requirement for $Il9$ CNS-25 for expression in Tc9 cells was assessed. WT and $Il9^{\Delta CNS-25}$ CD8$^+$ T cells cultured with TGFβ and IL-4 showed induction of IL-9 that was completely dependent upon CNS-25 (Fig. 3f, g, and Supplementary Fig. 3e). However, expression of $Ifng$ and other Tc-associated genes were not altered (Supplementary Fig. 3f–h).

IL-9 can also be produced by other Th subsets. To assess if CNS-25 deficiency similarly affects IL-9 in other subsets, naive CD4$^+$ T cells from WT and $Il9^{\Delta CNS-25}$ mice were differentiated under Th1, Th2, Th17, or Treg cell inducing conditions. Differentiation of other subsets was normal as Th1 and Treg

cell expression of $Ifng$ and $Tbx21$, or $Il10$ and $Foxp3$, was not altered by CNS-25 deficiency (Supplementary Fig. 3i–l). In Th1 cells we observed very low levels of IL-9 that were not altered by CNS-25 deficiency (Fig. 3d). We could not detect intracellular IL-9 in Treg cultures, but the limited amount of $Il9$ mRNA was dependent on CNS-25 (Fig. 3d). In Th17 cultures, the hallmark Th17 cytokine production was not affected but we observed a 90% decrease in IL-9 production in the absence of CNS-25, similar to what was observed in Th9 cells (Fig. 3h–j and Supplementary Fig. 3m–o). CNS-25 deficiency did not affect IL-4 production, but interestingly increased IL-9 production in Th2 cells (Fig. 3k–m and Supplementary Fig. 3p–r), suggesting that a Th2 transcription factor is actively repressing $Il9$ expression through the CNS-25 element. Together, these results suggest that CNS-25 specifically promotes $Il9$ expression in Th9 and Th17 cells, but mediates repression of $Il9$ in Th2 cells.

**$Il9$ locus function in the absence of CNS-25**. To further assess how CNS-25 deficiency affected the $Il9$ locus, we performed ChIP assays for histone modifications. We observed that in the absence of CNS-25, H3K4 tri-methylation and H3K27 acetylation were diminished at CNS-6 and the $Il9p$ (Fig. 4a). In contrast, H3K27 tri-methylation was not different at CNS-6 and the $Il9p$ but was increased at non-conserved sites upstream of the gene (Fig. 4a). We further observed a general pattern of decreased transcription factor binding at the $Il9p$ in the absence of CNS-25, with significant decreases in STAT5, STAT6, IRF4, BATF, and PU.1 (Fig. 4b).

We further tested the "openness" of the locus using the nuclease accessibility assay. Th9 cells were differentiated from WT and $Il9^{\Delta CNS-25}$ mice and isolated nuclei were treated with a nuclease cocktail. In cells that lacked CNS-25, accessibility at the $Il9p$ was significantly decreased without an effect at a control locus (Fig. 4c, d). These data suggest that in the absence of CNS-25, the $Il9$ locus has a more "closed" configuration in Th9 cells.

The ability of transcription factors such as IRF4 and BATF to activate the $Il9$ locus is dependent on the accessibility of the locus, as they do not activate $Il9$ in Th2 or Th17 cells[18]. To test the ability of IRF4 and BATF to transactivate the $Il9$ locus in the absence of CNS-25, Th9 cells differentiated from WT and $Il9^{\Delta CNS-25}$ mice were transduced with retroviruses ectopically expressing IRF4 or BATF. In the absence of CNS-25 the basal amount of IL-9 was decreased, consistent with previous data (Fig. 4e, f). Transduction of IRF4 or BATF into WT cells increased the amount of IL-9 production (Fig. 4e, f). Importantly, both BATF and IRF4 transduced into $Il9^{\Delta CNS-25}$ Th9 cells were also able to increase IL-9 production (Fig. 4e, f). In BATF-transduced cells, the amount observed was similar to that in control transduced WT cells (Fig. 4e, f). Moreover, while BATF and IRF4 transduction did not overcome the loss of CNS-25 in the absolute amount of IL-9 produced, the fold induction by

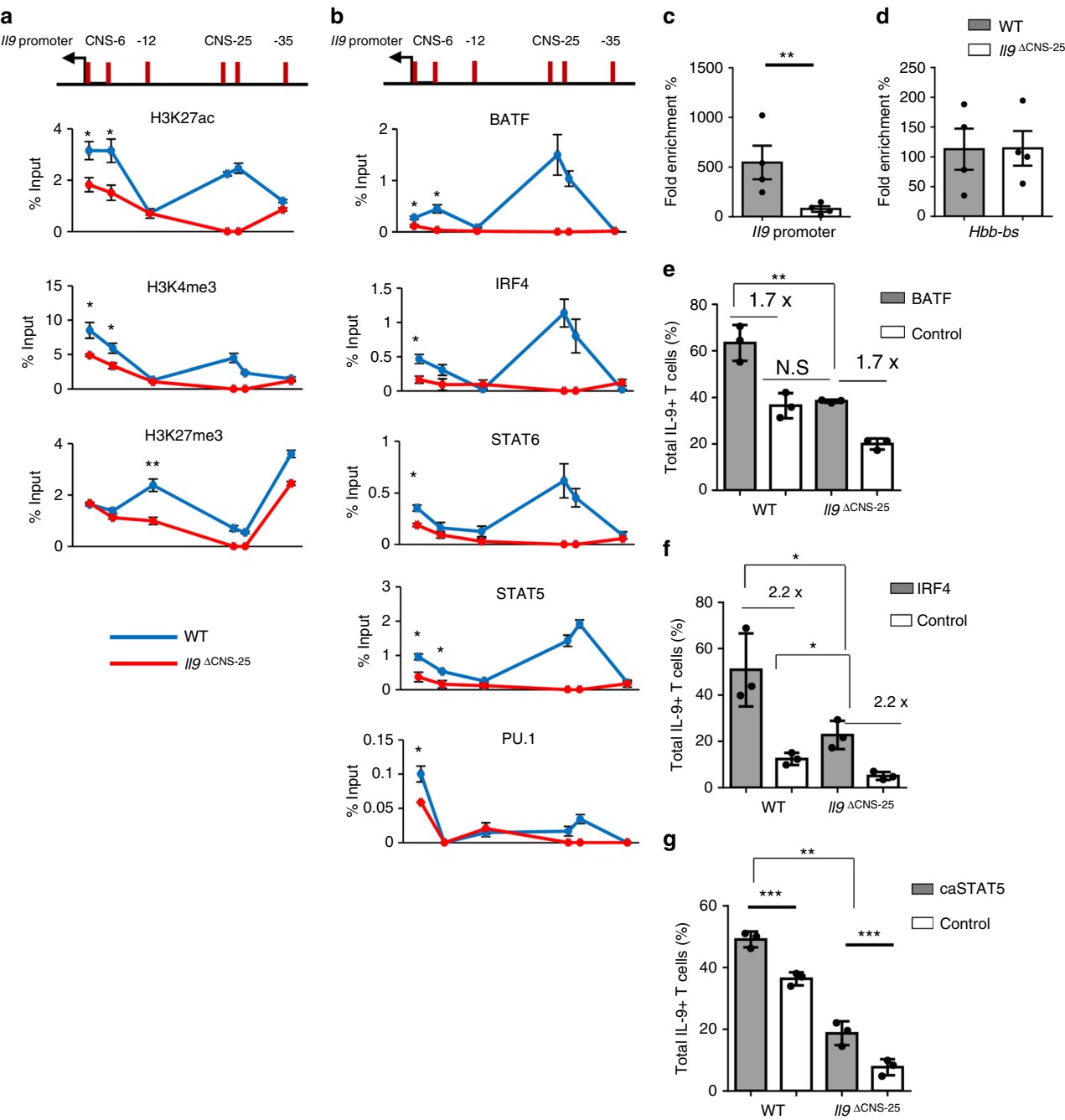

**Fig. 4** CNS-25 deletion alters chromatin at the *Il9* promoter in Th9 cells. Naive CD4$^+$ T cells were isolated from WT or *Il9*$^{\Delta CNS-25}$ mice and cultured under Th9 cell polarizing conditions. On day 5, cells were harvested for ChIP analysis (**a**, **b**) and chromatin accessibility assay (**c**, **d**). **a** Histone modifications at the *Il9* locus in Th9 cells. Modifications at the CNS-25 in *Il9*$^{\Delta CNS-25}$ T cells were not detected. **b** The binding of IL-9-inducing transcription factors at the *Il9* locus in Th9 cells. Transcription factor binding at the CNS-25 in *Il9*$^{\Delta CNS-25}$ T cells were not detected. **c**, **d** Fold enrichment was calculated by a ratio of qPCR Ct values of the nuclease-treated sample over the no-nuclease control sample in *Il9* promoter (**c**) and negative control region, *Hbb-bs* (**d**). **e–g** Retroviral transduction of IL-9-inducing TF in WT or *Il9*$^{\Delta CNS-25}$ Th9 cultures. Twenty-four hours after initiation of culture, activated T cells were transduced with BATF (**e**), IRF4 (**f**), or caSTAT5 (**g**) expressing retrovirus. On day 5, cells were restimulated with PMA/ionomycin for 6 h to measure cytokine production using intracellular staining. Data are mean ± SEM of 3 mice per group and representative of two independent experiments. A two-tailed Student's *t* test was used for pairwise comparisons. *$p < 0.05$, **$p < 0.01$, ***$p < 0.001$

either transcription factor was the same in WT and *Il9*$^{\Delta CNS-25}$ Th9 cells, suggesting that ectopic expression of the transcription factors can still activate the locus by binding to the promoter. Constitutively active STAT5 (caSTAT5) ectopic expression significantly increased IL-9 production in both WT and *Il9*$^{\Delta CNS-25}$ Th9 cells to a greater degree than IRF4 or BATF but IL-9 production was still not equal to WT transduced cells

(Fig. 4g). Thus, while all transcription factors tested are able to function through the *Il9* promoter and other remaining elements, overexpression does not overcome the contribution of the CNS-25 element.

***Il9* CNS-25 deficiency in innate immune cells.** A number of other cell types also express *Il9* including MCs, basophils, and

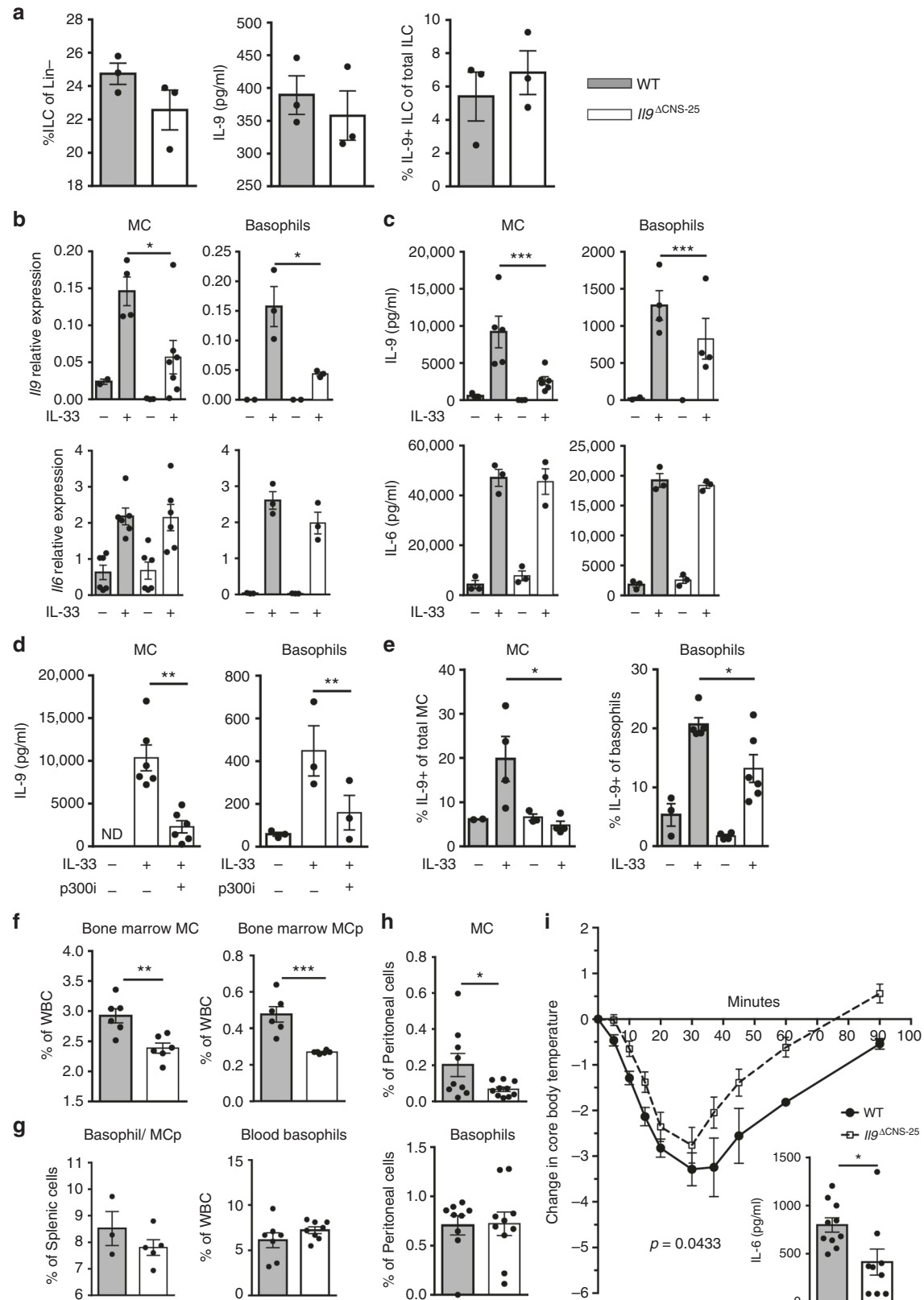

ILCs[48–51]. To determine if *Il9* CNS-25 was active in cells other than CD4[+] T cells, we examined IL-9 production in these populations in the absence of CNS-25. ILC2 are both IL-9 producers and IL-9 responsive, with homeostasis depending on IL-9[48,49]. To determine if ILC2 numbers or IL-9 production were affected by CNS-25 deficiency, WT and *Il9*[ΔCNS-25] mice were injected intranasally with IL-33 as previously described[48]. We observed no differences in either total ILC or ILC2 (Lin[−], CD45[+], KLRG1[+], Sca1[+], Thy1.2[+], and ST2[+]) accumulation in the lung and no difference in IL-9 production (Fig. 5a and Supplementary

**Fig. 5** CNS-25 deletion impairs IL-9 production in multiple lineages of IL-9-producing cells. **a** WT and $Il9^{\Delta CNS-25}$ mice were treated i.n. with 0.5 μg of IL-33 for 3 days. Lung ILCs were isolated and restimulated with IL-33. Frequency of ILCs in lung was analyzed by flow cytometry (left) and IL-9 production by ILCs was analyzed by ELISA (middle) and intracellular staining (right). **b, c** IL-9 and IL-6 mRNA (**b**) and protein (**c**) levels were assessed by qRT-PCR or ELISA, respectively, in IL-33-stimulated mast cells (MCs) and basophils derived from WT and $Il9^{\Delta CNS-25}$ bone marrow. **d** MC and basophils derived from WT and $Il9^{\Delta CNS-25}$ bone marrow were treated with p300i for 1 h prior to stimulation with IL-33. IL-9 production was assessed using ELISA. **e** WT and $Il9^{\Delta CNS-25}$ mice were treated i.n. with 0.5 μg of IL-33 for 3 days. Peritoneal cells (left) and white blood cells (WBC) (right) were restimulated with IL-33 and intracellular IL-9 production was measured in MC and basophils by flow cytometry. **f** MC (left) and mast cell precursor (MCp) (right) frequencies were assessed in bone marrow of WT and $Il9^{\Delta CNS-25}$ mice. **g** Basophil mast cell progenitor frequencies in spleen (left) and basophil frequencies in blood (right) were analyzed via flow cytometry in WT and $Il9^{\Delta CNS-25}$ mice. **h** Frequencies of mast cells (top) and basophils (bottom) in peritoneum of WT and $Il9^{\Delta CNS-25}$ mice. **i** WT and $Il9^{\Delta CNS-25}$ mice were injected with anti-DNP IgE (50 μg) 16 h before passive systemic anaphylaxis (PSA) ($n = 9$–10). PSA was induced by administering 100 μg of DNP-BSA. Core body temperature was measured over 90 min period and mice were sacrificed. Inset, after sacrifice, plasma IL-6 level was measured using ELISA ($n = 4$). AUC used to compare response in WT and $Il9^{\Delta CNS-25}$ mice ($p = 0.043$). Unless otherwise stated, all data are mean ± SEM and representative of at least two independent experiment done in triplicates and statistical significance assessed by using unpaired Student's $t$ test (*$p < 0.05$; **$p < 0.01$; ***$p < 0.001$). N.D. not detected

Fig. 4a, b). This suggests that $Il9$ CNS-25 is not required for ILC2 IL-9 production, and that the amount of IL-9 lost in $Il9^{\Delta CNS-25}$ mice is not sufficient to negatively affect ILC2 numbers in this system.

To determine if IL-9 production is dependent on CNS-25 in MCs and the related basophil cell population, we differentiated each cell type from bone marrow of WT and $Il9^{\Delta CNS-25}$ mice. Cell growth of MCs and basophils in culture was not altered by CNS-25 deficiency, and the modest increases in Fc receptors and CD49b expression on the surface of $Il9^{\Delta CNS-25}$ cells are unlikely to be biologically significant (Supplementary Fig. 4c–e). MCs and basophils from $Il9^{\Delta CNS-25}$ mice demonstrated diminished IL-9 production compared to WT cells with no changes in IL-6 production (Fig. 5b, c). Bone marrow-derived MC and basophil production of IL-9, like Th9 cells, was dependent on p300 (Fig. 5d). Peritoneal MCs and basophils were also isolated from $Il9^{\Delta CNS-25}$ mice treated with IL-33 as in Fig. 5a and observed to produce less IL-9 than WT cells (Fig. 5e). To further characterize the effects of $Il9$ CNS-25 deficiency on MC homeostasis we enumerated mature and precursor cells using established markers[52]. MC precursor numbers in the bone marrow and MC numbers in the bone marrow and peritoneal cavity were significantly decreased in $Il9^{\Delta CNS-25}$ mice (Fig. 5f, h). In contrast, basophil/MC precursors in the spleen, and mature basophils in the blood and peritoneal cavity of $Il9^{\Delta CNS-25}$ mice were indistinguishable from WT mice (Fig. 5g, h). To determine if the decrease in MC numbers had an effect on function, we performed a passive systemic anaphylaxis assay and observed that $Il9^{\Delta CNS-25}$ mice had significantly less temperature change (area under the curve, $p = 0.043$) and significantly less serum IL-6 at the termination of the assay (Fig. 5i). Together, these data indicate that $Il9$ CNS-25 is a critical regulator of IL-9 in T cells, MCs, and basophils, but not ILCs.

**Il9 CNS-25 deficiency in vivo.** To continue to test the effects of CNS-25-deficiency in vivo, WT and $Il9^{\Delta CNS-25}$ mice were exposed to *Aspergillus fumigatus* extract over a period of 3 weeks (Fig. 6a). Although there were no differences in the overall cell and eosinophil number in the bronchoalveolar lavage (BAL), analysis of $CD4^+$ T cells in the BAL and lung revealed that the proportion of IL-9-secreting T cells was significantly decreased in mice lacking CNS-25 (Fig. 6b–d). In contrast, there were no differences in the percentages of IL-17A-positive cells (Fig. 6d).

To further assess the effects of CNS-25 deficiency on the inflammatory process, we assessed additional parameters of allergic inflammation. Mucus production in the lung, as assessed by periodic acid–Schiff (PAS) staining of tissue and *Muc5ac* and *Clca3* expression, was significantly decreased in $Il9^{\Delta CNS-25}$ mice compared to WT mice (Fig. 6e, f). Consistent with mucus

production having a major contribution to airway reactivity in mice[53], $Il9^{\Delta CNS-25}$ mice had diminished methacholine reactivity compared to WT mice (Fig. 6g). Consistent with MC accumulation in allergic inflammation being dependent on IL-9 and Th9 cells[54,55], there were significantly diminished numbers of MCs in the trachea from $Il9^{\Delta CNS-25}$ mice, compared to WT mice (Fig. 6h). Together, these data indicate that $Il9$ CNS-25 controls IL-9 production in vivo and that CNS-25-dependent IL-9 controls mucus production, MC accumulation, and airway hyperreactivity.

**Function of the human *IL9* CNS.** By virtue of $Il9$ CNS-25 being a CNS, there is a homologous sequence 18 kb upstream of the human *IL9* gene (Fig. 7a). However, there is the possibility that the human and mouse genes encoding IL-9 might be regulated differently. Although *IL9* is linked (3.2 Mb away) to the *IL4-IL13-IL5* locus in the human genome on chromosome 5, $Il9$ is on a separate chromosome (chromosome 13) from the *Il4-Il13-Il5* locus (chromosome 11) in the mouse genome. Thus, to directly determine if *IL9* CNS-18 functioned similarly to $Il9$ CNS-25, we first performed ChIP assays for histone modifications. Naive $CD4^+CD45RA^+$ T cells were isolated as described in methods and the purity of naive T cells after magnetic separation was determined to define contamination from terminally differentiated effector memory cells re-expressing CD45RA (TEMRA). The purified population was over 95% $CD4^+CD45RA^+$ and the percentage of $KLRG1^+$ and/or $CD57^+$ cells, markers of TEMRA, was <1.5% (Supplementary Fig. 5b). After in vitro differentiation under polarizing conditions, we observed that *IL9* CNS-18 was enriched for monomethyl-H3K4 and acetyl-H3K27 in human Th9 cultures compared to Th0 cells (Fig. 7a). To more directly test the function of *IL9* CNS-18, we used a Cas9-EGFP fusion protein-encoding lentivirus that also expressed two guide RNAs targeting sequences surrounding CNS-18 to infect human Th9 cell cultures. Gating on the green fluorescent protein (GFP)-positive cells, we observed that deletion of CNS-18 reduced IL-9 production among multiple donors (Fig. 7b, c). PCR verified deletion of the CNS-18 sequences and the percent deletion correlated significantly with the percent decrease in *IL9* expression (Fig. 7d). Importantly, deletion of CNS-18 had no effect on the expression of cytokines; *IL10* and *IL21*, which are produced by Th9 cells (Fig. 7e) or *IL5*, *IL13*, and *IL3*, which are linked to same chromosome with *IL9* (Supplementary Fig. 5a) in transduced cells. *IL4* expression was not detected. Thus, *IL9* CNS-18 is an important regulatory element of the *IL9* gene in human Th9 cells.

**Discussion**
Although IL-9 can be produced by many cell types within the context of pathogen immunity, tumor immunity, and

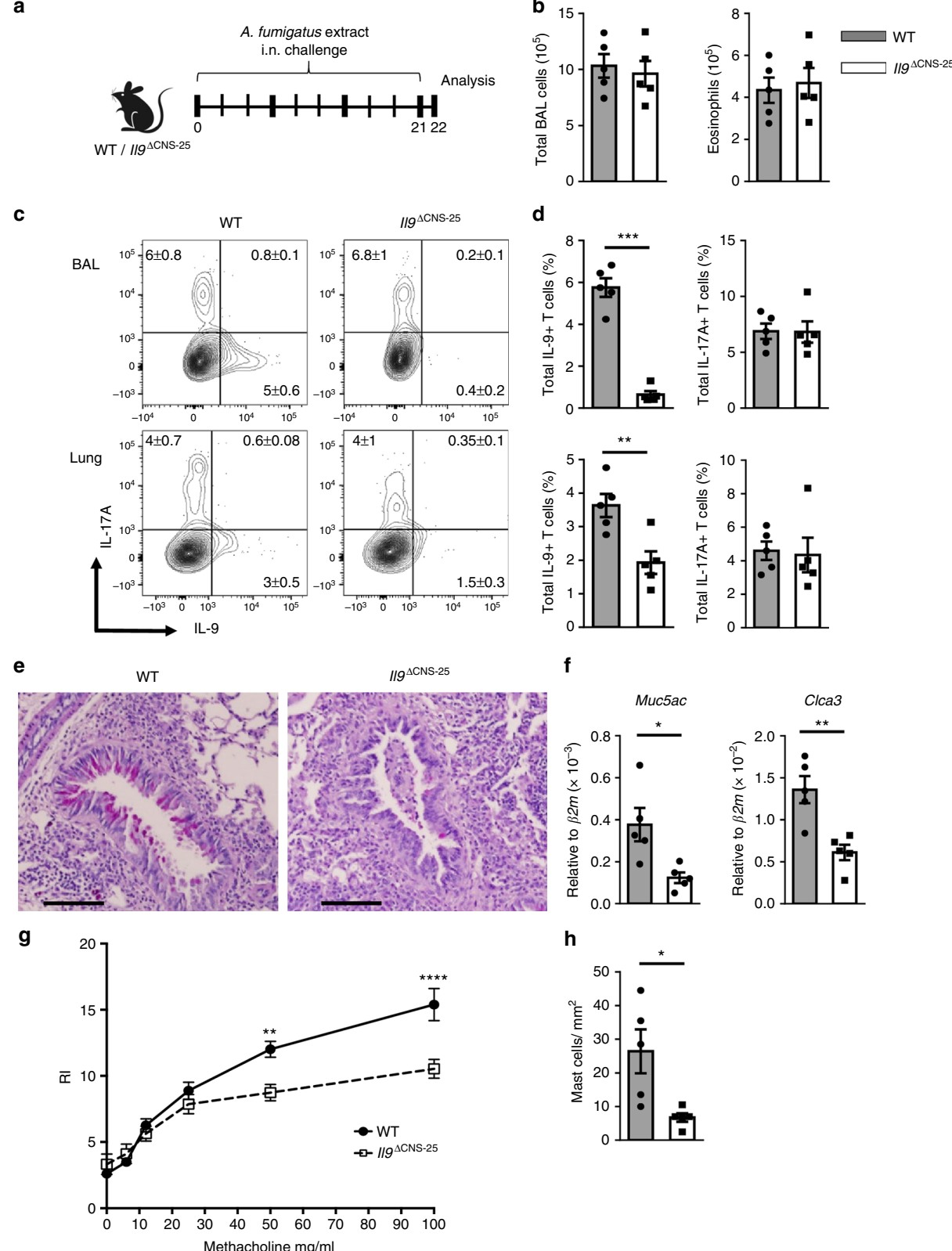

inflammatory disease, the regulation of this cytokine is still largely uncharacterized. In this report we have defined an *Il9* regulatory element, CNS-25, that has all the hallmarks of an enhancer and is required for normal IL-9 production in multiple cell types including T cells, MCs, and basophils. Importantly, the function

of this element is conserved in human T cells where the *IL9* CNS-18 is required for optimal IL-9 production in human Th9 cells.

The *Il9* CNS-25 element confers multiple levels of specificity to the IL-9 gene. Within both human and mouse Th9 cells, CNS-25/-18 promoted IL-9 expression under standard culture

**Fig. 6** *Il9*$^{\Delta CNS-25}$ mice have attenuated IL-9 production and airway reactivity in *A. fumigatus* extract-induced airway inflammation. WT and *Il9*$^{\Delta CNS-25}$ mice were intranasally challenged with 25 μg of *A. fumigatus* extract 3 times a week for 21 days. **a** Schematic of *A. fumigatus* extract-induced airway inflammation model. **b** Total cell and eosinophil numbers in the BAL from mice harvested on day 22. **c**, **d** BAL and lung cells were restimulated with PMA/ionomycin for 6 h to measure cytokine production using intracellular staining. Representative dot plots (**c**) and average percentages (**d**) of IL-9-producing or IL-17-producing CD4$^+$ T cells in BAL (top) and lung (bottom). Cells for flow cytometric analysis were gated on lymphocyte size and granularity, and the expression of CD4 and TCRβ. Data are mean ± SEM of 5 mice per group. A two-tailed Student's *t* test was used for pairwise comparisons. *$p < 0.05$, **$p < 0.01$, ***$p < 0.001$. **e** Representative images of PAS staining of inflamed lung tissue. Scale bars = 100 μm. **f** *Muc5ac* and *Clca3* expression in the lung tissues. Gene expression were normalized to *β2m* expression. **g** WT ($n = 7$) and *Il9*$^{\Delta CNS-25}$ ($n = 9$) mice were treated intranasally with 25 μg *A. fumigatus* extract three times a week for 3 weeks. RI was measured in response to methacholine challenge 2 h after the last intranasal antigen treatment. Data are mean ± SEM of WT ($n = 7$) and *Il9*$^{\Delta CNS-25}$ ($n = 9$). One-way ANOVA with a post hoc Tukey test was used to generate *p* values for multiple comparisons (**$p < 0.0021$, ****$p < 0.0001$). **h** Number of mast cells in trachea after Toluidine blue staining of trachea counted from two fields of each specimen and averaged. **d**, **f**, **h** Data are mean ± SEM of 5 mice per group and representative of two independent experiments. A two-tailed Student's *t* test was used for pairwise comparisons. *$p < 0.05$, **$p < 0.01$, ***$p < 0.001$

conditions, and for mice, IL-9 production in vivo as well. *Il9* CNS-25 also promoted IL-9 in MCs, basophils, Th17 cells, Tc9 cells, but surprisingly not in ILCs. The lack of an effect of CNS-25 deletion in ILCs on IL-9 production was observed in cells isolated from IL-33-treated-mice. This would suggest that *Il9* regulation in ILCs is intrinsically different from other immune cells. Previous studies have suggested that ILC cytokine loci are primed for expression to a greater degree than in T cells[56,57]. Indeed, the *Il9* locus had more ATAC-seq peaks in precursor and mature ILC2 than Th2 cells[57], and the ATAC-seq profile in ILCs corresponds to the p300 ChIP-seq profile in Th9 cells. It is possible that other enhancers can compensate for the loss of *Il9* CNS-25 in ILCs, a common concept in development where disruption of multiple regulatory elements is needed for a phenotypic change[58]. Regardless of the differences in control between ILCs and other immune cells, the lack of an effect in ILCs further demonstrates the specificity of the effects of *Il9* CNS-25 deletion.

Also striking was the repressor effect observed on the loss of *Il9* CNS-25 in Th2 cells where IL-9 production was increased in *Il9*$^{\Delta CNS-25}$ cultures, compared to WT cultures. The amount of IL-9 produced by *Il9*$^{\Delta CNS-25}$ Th2 cells is still far less than observed from Th9 cultures, suggesting that CNS-25 is not the sole factor controlling the distinction in IL-9 production between these cell types. Additional factors are clearly promoting *Il9* expression in Th9 cells to a greater degree. However, it does suggest that there is a transcription factor in Th2 cells, not expressed in Th9 cells, that binds to CNS-25 and attenuates IL-9 production. While the identity of this transcription factor is not yet clear, it mirrors one role of PU.1, which represses Th2 cytokines as it induces IL-9 production[16,59,60]. Further studies will help to define this transcription factor and its importance in the balance between Th2 and Th9 cell differentiation.

There are some parallels between these observations and the recent descriptions of super-enhancers (SEs) at the *Il9* locus[29]. That report found three SE regions, defined with broad peaks of acetylated H3K27 and Brd4 binding that seemed to roughly overlap with CNS + 5.5, CNS-6, and CNS-25. Xiao et al.[29] found that the CNS + 5.5 region mediated the OX40-stimulated IL-9 production and observed little effect on OX40-stimulated IL-9 when a region encompassing CNS-25 was deleted. This likely represents differences between the response elements to OX40 and antigen receptor stimulation and that basal IL-9 production, in the absence of OX40 stimulation, was low in those studies. Thus, from the published studies it is impossible to determine if the region 3-prime of the *Il9* gene is important for induction of IL-9 to other stimuli, or if it controls expression of *Il9* in other cell types. The importance of the CNS + 5.5 region is interesting in that it is the least conserved of the CNS regions in the *Il9* locus, being conserved between mouse and humans, but not extensively with other species[28]. It would be worth exploring whether

OX40 stimulation of IL-9 was restricted to certain species, or if it activated the expression through other elements in the *Il9* locus in other species.

*Il9* CNS-25 clearly contributes to the overall structure of the *Il9* locus. We observed that *Il9* CNS-25 is physically linked to the *Il9* promoter in a 3C assay. This observation is supported by multiple proteins indicative of higher-order locus structure including CTCF, Smc1a, and Mediator complex components that were bound specifically in Th9 cells at the *Il9p* and CNS-25. It is also possible that deletion of the *Il9* CNS-25 element results in changes to the higher-order structure of the locus, perhaps also preventing the interaction of other elements with the *Il9p*. In that respect, some effects observed in these studies might be secondary to the impact of CNS-25 on other elements. Still, our results point to a central role for CNS-25 in establishing IL-9 production.

There is also the possibility that deleting cytokine regulatory elements might affect IL-9 production by feedback effects. In mouse cells, it is possible that IL-9 might feedback to control its own production. However, in experiments that block or supplement IL-9 during in vitro differentiation, we have not observed any changes in IL-9 production. In human cells, the *IL9* CNS-18 is physically linked to other Th2 cytokines and this might represent another potential indirect effect of deletion of the element. However, we detected no *IL4* expression in human Th9 cultures, and no significant change in the expression of *IL5*, *IL13*, or *IL3* upon *IL9* CNS-18 deletion, suggesting that this does not explain the decreases in IL-9 production in human Th9 cells when CNS-18 is deleted. Together, these data support a role for this element in specifically regulating the *Il9/IL9* gene.

*Il9* CNS-25 binds most of the transcription factors known to impact IL-9 production[16–22,26,61]. It is interesting that PU.1 is unique in binding predominantly to the *Il9* promoter perhaps suggesting that it is one of the initial activating transcription factors. The ability of *Il9* CNS-25 to bind multiple transcription factors suggests that there might be cooperative functions of the transcription factors. Indeed, in examining IRF4-deficient and STAT6-deficient Th9 cells, we see decreases in other transcription factors bound to the CNS-25 region and decreases in modifications associated with activated chromatin. We saw similar cooperative binding between BATF and IRF4 at the *Il9* promoter, and others have seen cooperative binding at the promoter with IRF4 and Smad proteins[25,62]. Further studies will more completely define these cooperative interactions in detail.

Although *Il9* CNS-25 identified in our studies is clearly important for IL-9 production, it does not negate the importance of other elements. The *Il9* CNS-6 clearly binds some IL-9-inducing transcription factors, is a p300-binding peak, and has enhancer activity in a reporter assay. The *Il9* CNS + 5.5 region is clearly required for IL-9 induction by OX40 stimulation[29]. In the p300 ChIP-seq data, we also observed a p300-binding peak at

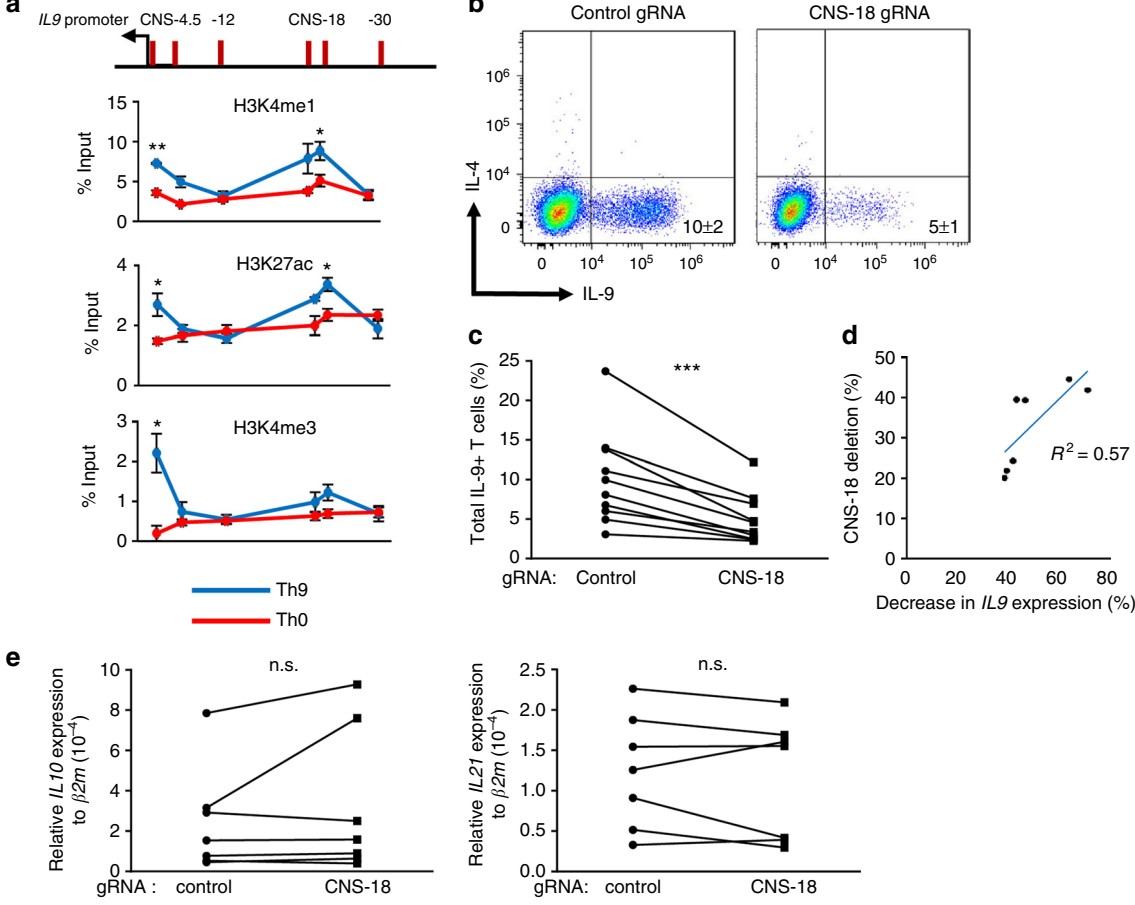

**Fig. 7** *IL9* CNS-18 is important for IL-9 production in human Th9 cells. **a** Schematic CNSs in the human *IL9* locus (top). ChIP assay of histone modifications at the *IL9* locus in human Th0 and Th9 cells. Naive CD4+ T cells were isolated from human PBMCs and cultured under Th0 and Th9 cell polarizing conditions. On day 4, cells were harvested for ChIP analysis. Data are mean ± SEM of three donors per group and representative of two independent experiments. A two-tailed Student's *t* test was used for pairwise comparisons. *$p < 0.05$, **$p < 0.01$. **b–e** Naive CD4+ T cells were isolated from human PBMCs and cultured under Th9 cell polarizing conditions. After 1 day of culture, cells were transduced with lentivirus expressing Cas9 and gRNAs targeting CNS-18 or a mouse genomic sequence as a control. **b, c** On day 4, cells were restimulated with PMA/ionomycin for 6 h to assess IL-9 production using intracellular staining (**b**), with results indicated for multiple donors (**c**). Data are mean ± SEM of 10 donors per group and cumulative from two independent experiments. A ratio paired *t* test was used for pairwise comparisons. ***$p < 0.001$. **d, e** On day 4, Cas9 and gRNAs expressing Th9 cells were sorted based on EGFP expression to measure cytokine expression and CNS-18 deletion efficiency through qPCR. **d** The graph indicates the percent decrease in PCR of the CNS-18 sequence versus the percent decrease in *IL9* mRNA expression. The *IL9* expression and deletion efficiency of *IL9* CNS-18 were normalized to *β2m* expression and amplification in −12 kb sequence respectively. $R^2 = 0.57$ and *$p < 0.05$. Data are mean ± SEM of seven donors. **e** qRT-PCR analysis of *IL10* and *IL21* in Th9 cells transduced with control or CNS-18-targeting lentivirus. n.s. not significant

about −20 kb from the *Il9* TSS. Although this region is not evolutionarily conserved, it might still contribute to *Il9* regulation and perhaps similar elements can be identified in other species. We did look across a larger window (±100 kb) on either side of the *Il9* gene and did not see other Th9 p300 peaks or peaks from other Th subsets that identified other potential regulatory elements. How all of these elements function in concert remains to be further explored.

There were multiple effects of diminished IL-9 in vivo, although IL-9 production in Th cells, MCs, and basophils was not completely ablated, and IL-9 production appeared normal in ILCs. In a passive systemic anaphylaxis model, *Il9*ΔCNS-25 mice showed less severe anaphylactic responses and serum cytokine responses. This could be due both to diminished MC homeostasis as well as effects of IL-9 directly on mature MCs and basophils. In the *Aspergillus* extract model of allergic airway disease *Il9*ΔCNS-25 mice demonstrated attenuate mucus production, decreased

airway reactivity, and decreased MC accumulation in the trachea. In contrast, there was no decrease in the overall infiltration into the airways or IL-17-secreting CD4 T cells in the lung, suggesting that the observations were due to specific effects of IL-9 in allergic inflammation and not because of overall diminished inflammation. Thus, the *Il9*ΔCNS-25 mice present a useful model to further explore IL-9-specific responses in allergic disease.

IL-9 is a critical cytokine in immunity to helminthic parasites, tumor immunity, and in inflammatory diseases including autoimmunity and allergic inflammation. Understanding the regulation of IL-9 will allow manipulation of signals and transcription factors that can control IL-9 production to alter the course of disease, and potentially amplify the ability of Th9 cells to mediate beneficial immunity. The identification of *Il9* CNS-25 as a centrally important control element of the IL-9 gene in multiple cell types, and the finding that function of this element is conserved in human T cells, suggests that these findings represent an entry

point to an ability to control IL-9 production in the context of immunity and disease.

## Methods

**Mice.** All mice were C57BL/6 background. CNS-25-deleted mice ($Il9^{\Delta CNS-25}$) were generated by CRISPR/Cas9-mediated gene editing (Taconic, Germany) (Supplementary Table 1). Mice were maintained under specific pathogen-free conditions. Experiments were performed with $Il9^{\Delta CNS-25}$ mice and their littermate controls. Other experiments were done with C57BL/6 female and/or male mice purchased from the Jackson Laboratory. All experiment were done with 6- to 12-week old mice. All experiments were performed with the approval of the Indiana University Institutional Animal Care and Use Committee.

**Human.** De-identified peripheral blood mononuclear cells (PBMCs) from buffy coat blood packs were obtained from the Indiana Blood Center. Studies were approved by the Institutional Review Board.

**In vitro mouse T-cell differentiation.** Naive CD4$^+$CD62L$^+$ T cells from mice were positively selected from the enriched CD4$^+$ T cells from spleen and lymph nodes using MACS beads and columns (Miltenyi Biotec). Naive CD4$^+$CD62L$^+$ T cells were activated with plate-bound anti-CD3 (2 Units/ml$^{-1}$ 145-2C11; BioXCell) and soluble anti-CD28 (0.5 μg ml$^{-1}$; BD Pharmingen) in complete culture media (cRPMI), Roswell Park Memorial Institute 1640 (RPMI 1640, ThermoFisher Scientific) containing 10% fetal bovine serum (FBS, Atlanta Biologicals), 1% antibiotics (penicillin and streptomycin/stock; Pen 5000 Units ml$^{-1}$, Strep 5000 μg ml$^{-1}$), 1 mM sodium pyruvate, 1 mM L-glutamine, 2.5 ml of non-essential amino acids (stock; 100×), 5 mM HEPES (all from LONZA), and 57.2 μM 2-mercapoethanol (Sigma-Aldrich), to generate Th0 cells (10 U hIL-2) or with additional cytokines (all from PeproTech) and cytokines (all from BioXCell) to generate Th2 cells (20 ng ml$^{-1}$ IL-4; and 10 μg ml$^{-1}$ anti-interferon gamma (IFNγ) XMG), Th9 cells (20 ng ml$^{-1}$ IL-4, 2 ng ml$^{-1}$ hTGF-β1, and 10 μg ml$^{-1}$ anti-IFNγ XMG), Treg cells (5 ng ml$^{-1}$ hTGF-β1, 10 U hIL-2, 10 μg ml$^{-1}$ anti-IFNγ, and 10 μg ml$^{-1}$ anti-IL-4), Th17 cells (100 ng ml$^{-1}$ IL-6, 2 ng ml$^{-1}$ hTGF-β1, 10 μg ml$^{-1}$ anti-IFNγ, and 10 μg ml$^{-1}$ anti-IL-4), and Th1 cells (20 ng ml$^{-1}$ IL-12, 10 U hIL-2, and 10 μg ml$^{-1}$ anti-IL-4 11B11) culture conditions. Cells were grown at 37 °C under 5% CO$_2$ and were expanded after 3 days with original concentration of cytokines in fresh medium. For p300i studies, cells were treated with 20 nM of p300i (SGC-CBP30, TOCRIS) on day 3. Cells were harvested on day 4 or 5 for analysis.

CD8$^+$ T cells from mice were positively selected using CD8a (Ly-2) microbeads (Miltenyi Biotec). CD8$^+$ T cells were activated with plate-bound anti-CD3 (2 μg ml$^{-1}$ 145-2C11; BioXCell) and soluble anti-CD28 (0.5 μg ml$^{-1}$; BD Pharmingen) in cRPMI to generate Tc9 cells (20 ng ml$^{-1}$ IL-4, 2 ng ml$^{-1}$ hTGF-β1, and 10 μg ml$^{-1}$ anti-IFNγ XMG). Cells were grown at 37 °C under 5% CO$_2$ and harvested on day 2 for analysis.

**In vitro human T-cell isolation and differentiation.** PBMCs were isolated from de-identified buffy coat blood packs (Indiana Blood Center, IN) by density gradient centrifugation using Ficoll-Paque (GE Healthcare). Human naive CD4$^+$CD45RA$^+$ T cells were isolated from human PBMCs using magnetic separation (Miltenyi Biotec). These cells were activated with an equal ratio of receptor crosslinking bead, Dynabead human T-activator CD3/CD28 (ThermoFisher Scientific) in cRPMI to generate Th0 cells (10 U hIL-2) and Th9 cells (20 ng ml$^{-1}$ hIL-4, 2 ng ml$^{-1}$ hTGF-β1, and 10 μg ml$^{-1}$ anti-IFNγ), and grown at 37 °C under 5% CO$_2$. Cells were harvested on day 4 or 5 for analysis.

**In vitro MC and basophil generation.** To generate MCs and basophils, bone marrow cells from WT and $Il9^{\Delta CNS-25}$ mice were isolated and red blood cells (RBCs) were lysed using Ammonium-Chloride-Potassium (ACK) lysis buffer (LONZA). Cells were cultured in cRPMI with IL-3 (10 ng ml$^{-1}$) and stem cell factor (SCF; 30 ng ml$^{-1}$) (all from BioLegend) for MCs or 20 ng ml$^{-1}$ of IL-3 for basophils. After 7 days of culture, basophils were isolated using anti-PE CD49b microbeads and MCs were allowed to mature for 21 days. Mature MCs and basophils were set to 1 × 10$^6$ cells per ml in 10 ng ml$^{-1}$ of both IL-3 and SCF and stimulated with 50 ng ml$^{-1}$ of IL-33 (BioLegend) for 4 h (quantitative real-time PCR; qPCR) or 16 h (ELISA). For p300i work, MCs and basophils were cultured as stated above and were treated with 20 nM of p300i (SGC-CBP30, TOCRIS) 1 h before stimulation with 50 ng ml$^{-1}$ of IL-33 for 16 h.

**Innate cell isolation.** For lung ILC studies, WT and $Il9^{\Delta CNS-25}$ mice were treated intranasally (i.n.) with 0.5 μg of IL-33 for 3 days. For single-cell suspension, lung tissues were minced and incubated with 0.5 mg ml$^{-1}$ of collagenase A (Roche) in Dulbecco's modified Eagle's medium (DMEM) at 37 °C for 45 min. After grinding tissues with mesh stainless steel strainer, RBCs were removed by ACK lysing buffer (LONZA) and reaction was stopped by adding buffer (phosphate-buffered saline (PBS) with 0.5% bovine serum albumin (BSA)), cells were washed with the buffer followed by filtering through 70 μm nylon mesh to remove debris. Lung cell suspension was lineage-depleted using lineage depletion kit and biotinylated

antibodies (lineage depletion kit and CD11c, CD19, DX5, and NK1.1) (all from Miltenyi Biotec). For intracellular IL-9 measurement, the lineage-negative cells were restimulated with 50 ng ml$^{-1}$ of IL-33 for 5 h and treated with monensin for 2 h. These cells were stained with Thy1.2 and IL-9. For detection of intracellular IL-9, these cells were selected as viability dye-negative and Thy1.2-positive via flow cytometry. For ELISA, Thy1.2-positive cells were selected as follows. Cells were stained with Thy1.2-PE and isolated using anti-PE beads according to the manufacturer's instruction (Miltenyi Biotec). Cells were restimulated with 50 ng ml$^{-1}$ of IL-33 in cRPMI for 16 h.

For peritoneal cell isolation, WT and $Il9^{\Delta CNS-25}$ mice were treated i.n. with 0.5 μg of IL-33 for 3 days. Peritoneal cells were isolated by washing the peritoneal cavity using PBS. White blood cells (WBC) were isolated by collecting blood from animals using cardia puncture. Blood RBCs were lysed using ACK lysis buffer (LONZA). Peritoneal cells and WBC were cultured in cRPMI with IL-3 and SCF supplement (10 ng ml$^{-1}$ for both). Cells were restimulated immediately with 50 ng ml$^{-1}$ of IL-33 and intracellular IL-9 production was measured in MC and basophils via flow cytometry. MCs were gated as FcεR1 and c-kit double-positive and basophils were gated as FcεR1 and CD49b double-positive.

To determine basophil and MC frequencies, bone marrow cells and peritoneal cells were isolated as described above. The frequencies of basophils and MCs and their progenitors were gated as follows. Mature MCs in bone marrow and peritoneum were gated as FcεR1 and c-kit double-positive. Mature basophils in blood and peritoneum were gated as FcεR1 and CD49b double-positive. Bone marrow MC progenitors (MCps) were gated as lineage-negative (CD5, B220, CD11b, CD27, anti-Gr-1(Ly6G/C), Ly6C, Sca1, Ter119, CD19, and NK1.1) FcεR1$^{low}$c-Kit$^+$ST2$^+$β7$^+$. Basophil MCp in the spleen was gated as lineage-negative (B220, CD3, Ly6C/G, NK1.1, GR1, Ter119, and CD5) c-Kit$^+$FcγRII/III$^{hi}$β7$^{hi}$ST2$^+$. Basophils and MC and their progenitor frequencies were identified using previously published surface marker criteria[63].

**Retrovirus production and transduction.** Platinum E cells were grown in 10 ml of DMEM 1640 with 10% FBS and 1% antibiotics in a 100 mm tissue culture dish. When confluency reached 80–90%, cells were transfected with control vector or retroviral vector containing BATF, IRF4, or caSTAT5 open reading frame using Lipofectamine 3000 (ThermoFisher Scientific) according to the manufacturer's instructions. After 24 h of transfection, the media was collected for retroviral transduction or stored at −80 °C for subsequent use.

Activated mouse CD4$^+$ T cells were infected on day 1 with retrovirus containing control or expressing the interested gene by centrifugation at 2300 rpm at 32 °C for 90 min in the presence of 8 μg ml$^{-1}$ polybrene (Sigma-Aldrich). After spin infection, the supernatant was replaced with the fresh Th cell condition media. Cells were expanded on day 3 and analyzed on day 4 or 5.

**CRISPR/Cas9 plasmid construct.** PX330A_D10A-1 × 2 (Addgene #58772) was modified by adding ClaI site in front of hU6 promoter, termed as "new pX330A_D10A-1 × 2". gRNAs were designed using the Zhang lab's online tool (http://crispr.mit.edu/). After annealing of gRNA oligos, the gRNA duplexes were cloned to modified pX330A_D10A-1 × 2 and pX330S-2 (Addgene #58778) using BbsI (BpiI) (Supplementary Table 2). Through Golden gate assembly using Eco31I, the modified pX330A_D10A-1 × 2, which contains gRNA cassette containing two hU6 promoter and two gRNAs was generated. By using Cla1 and Kpn1, the gRNA cassette from this vector was inserted to new lentiCRISPR v2 (modified from lentiCRISPR v2, Addgene #52961, by adding new cloning site containing Cla1 and Kpn1) using Cla1 and Kpn1. Finally, a DNA element containing gRNA cassette of this new lentiCRISPR v2 was replaced with lentiCas9-EGFP (Addgene #63592) using Not1 and Nhe1. Plasmids and gRNA sequences are listed in Supplementary Table 2 and 3.

**Lentivirus production and infection.** Human embryonic kidney 293T cells were grown with 10 ml of DMEM 1640 with 10% FBS and 1% antibiotics in a 100 mm tissue culture dish. When confluency reached 95–99%, cells were transfected with lentiviral vectors expressing Cas9 and gRNAs targeting hIL9 CNS-18 or GM38602 as a negative control using Lipofectamine 3000. After 24 h of transfection, the virus was collected and concentrated with Lenti-X-concentrator (Takara Bio) according to the manufacturer's instructions.

Day 1 cultured human T-cells were infected by the lentivirus using Retronectin (Takara Bio) according to the manufacturer's instructions. Cells were harvested on day 4 or 5 for analysis using flow cytometry or sorted by fluorescence-activated cell sorter (FACS; BD FACSAria) based on enhanced GFP (EGFP) for gene expression analysis.

**A. fumigatus extract-induced allergic airway inflammation.** Mice were challenged intranasally with A. fumigatus (Greer Laboratories) extract 3 days a week for 21 days. A. fumigatus (25 μg) extract was diluted with PBS (20 μl) and administered into the nose. Mice were sacrificed 1 day after final intranasal challenge. BAL cells were collected by lavaging the lungs with 1 ml PBS. BAL and lung single-cell suspensions were stimulated with phorbol 12-myristate 13-acetate (PMA) and ionomycin for 6 h with monensin added after 2 h to assess cytokine production using intracellular staining. For ILC2 analysis, total lung cells were stained with

lineage-positive staining cocktail (BD Pharmingen), CD45, KLRG1, Sca1, Thy1.2, and ST2.

For airway resistance, 1 day after final intranasal challenge, mice were subjected to increased doses of methacholine and airway resistance was measured using plethysmograph chambers with a ventilator (Elan Series Mouse RC Site; Buxco Electronics) and BioSystem XA software (Buxco Electronics).

For histology, lungs and trachea were excised from the thoracic cavity followed by fixation with 4% neutral-buffered formaldehyde (ThermoFisher Scientific). Tissues were embedded in paraffin, sectioned, and stained with the PAS for lung tissue or toluidine blue staining for trachea.

**Passive systemic anaphylaxis.** For anaphylaxis experiment WT and $Il9^{\Delta CNS\text{-}25}$ mice were intraperitoneally (i.p.) injected with 200 μg of anti-DNP IgE (provided by Dr. Basar Bilgicer, Notre dame University) 16 h before they were i.p. challenged with 50 μg of DNP-BSA (Sigma-Aldrich). Core body temperature of animals was measured for 90 min using rectal thermometer (Braintree Scientific, Inc.) at which point plasma was collected via cardiac puncture to measure cytokine levels via ELISA.

**Quantitative real-time PCR.** Total RNA was extracted using TRIzol reagent (ThermoFisher Scientific) and reverse transcribed using cloned Avian Myeloblastosis Virus reverse transcriptase (ThermoFisher Scientific). For qPCR, Taqman real time PCR assay (ThermoFisher Scientific) or SYBR green master mix (Applied Biosystems) was used for gene expression analysis. Gene expression was normalized to housekeeping gene expression (β2-microglobulin). The relative gene expression was calculated by the change-in-threshold ($2^{-\Delta CT}$) method. All experiments were performed in duplicate in two independent experiments and results are presented as standard error of means of biological replicates. Taqman probes and eRNA primer sequences are listed in Supplementary Table 4 and 5.

**Flow cytometric analysis.** For cytokine staining, in vitro-cultured $CD4^+$ T cells were stimulated with PMA (Sigma-Aldrich) and ionomycin (EMD Millipore) for 2 h followed by monensin (BioLegend) for a total of 6 h. After fixation with 4% formaldehyde for 10 min at room temperature (RT), cells were washed two times with FACS buffer (PBS with 0.5% BSA). For transcription factor staining, cells were fixed with Foxp3/Transcription factor fixation buffer (eBioscience) at 4 °C in dark for 30 min or overnight. Fixed cells were permeabilized with permeabilization buffer (eBioscience), and stained for cytokines and transcription factors with fluorochrome-conjugated antibodies (1:200 dilution) at 4 °C in dark for 1 h. Stained cells were washed two times with FACS buffer and resuspended with 500 μl of FACS buffer for flow analysis. Fluorescent antibodies for flow cytometric analysis are listed in Supplementary Table 6.

**Enzyme-linked immunosorbent assay.** Cytokine capturing antibodies were coated on 96-well plate (NUNC) with coating buffer ($dH_2O$ with 0.1 M $NaHCO_3$ pH 9) and incubated at 4 °C overnight. After blocking the coated plate for 1 h with ELISA buffer (PBS with 2% BSA), diluted samples (10- to 100-fold diluted with ELISA buffer) and standard cytokines were added and incubated at 4 °C overnight. After washing, the plate was incubated with biotinylated secondary antibody at RT for 1 h followed by incubation with avidin-alkaline phosphatase (Sigma) at RT for 1 h. After washing the plate, $p$-Nitrophenyl Phosphate (Sigma-Aldrich) was added and measured with the Biorad Microplate 680 ELISA reader. Primary and secondary antibodies for ELISA are listed in Supplementary Table 7. For IL-6 and IL-9, ELISA kits (BioLegend) were used.

**Chromatin immunoprecipitation.** In vitro-differentiated Th cells were crosslinked for 15 min with 1% formaldehyde at RT with rotation. The reaction was quenched by adding 0.125 M glycine and incubated at RT for 5 min. Fixed cells were lysed with cell lysis buffer, followed by nuclear lysis buffer. Nuclei were degraded and chromosomal DNA were fragmented to a size range of 200–500 bp through ultrasonic processor (Vibra-cell). After sonication, the supernatant was diluted 10 fold with ChIP dilution buffer. After pre-clearing, the supernatant was incubated with the ChIP antibodies at 4 °C overnight with rotation. The following day, immunocomplexes were precipitated with Protein Agarose A or G beads at 4 °C for 2–4 h. Immunocomplexes were washed with low salt, high salt, LiCl and two times with TE buffer. After elution followed by reverse crosslinks, DNA was purified and analyzed by qPCR. After normalization to the Input DNA, the amount of output DNA of each target protein was calculated by subtracting that of the IgG control. ChIP antibodies and primer sequences are listed in Supplementary Table 8 and 9.

**Chromosome conformation capture assay.** In vitro-differentiated Th cells were crosslinked for 15 min with 1% formaldehyde at RT with rotation. The reaction was quenched by adding 0.125 M glycine and incubated at RT for 5 min. Fixed cells were lysed with ice-cold cell lysis buffer 10 mM Tris-HCl [pH 8], 10 mM NaCl, 0.2% NP-40, and protease inhibitor mixture). Nuclei were incubated with restriction enzyme buffer (NEB2) containing 0.3% SDS for 1 h while shaking at 900 rpm at 37 °C. After adding 2% Triton X-100, nuclei were incubated another 1 h. DNA was digested by adding 400 U Nco1 for overnight at 37 °C. Digestion was quenched

by adding 1.6% SDS for 25 min while shaking 900 rpm at 65 °C. After incubation of digested nuclei with T4 DNA ligase buffer containing 1% Triton X-100 for 25 min while shaking 900 rpm, DNA was ligated by adding 100 U T4 DNA ligase and incubated for 4 h at 16 °C followed by cooling to RT. DNA was reverse-crosslinked by adding proteinase K (10 mg $ml^{-1}$) and incubated for overnight at 65 °C. After incubation of DNA with RNase (10 mg $ml^{-1}$) for 45 min at 37 °C, DNA was purified and analyzed by qPCR. BAC clone (RP24-366P8) covering $Il9$ locus was digested and ligated for reference standard curve of 3C qPCR. Interaction frequencies were normalized to internal control.

**Luciferase reporter assay.** EL4 cells ($2 \times 10^6$ cells) or primary cultured T cells ($5 \times 10^6$ cells) were co-transfected with 5 μg of the $Il9$ gene locus containing pGL3 basic vector and 5 μg control or IRF4 expressing pcDNA3.1 vector and 0.5 μg of pRL-TK for endogenous control using Amaxa Nucleofection Kit L (Lonza). After 24 h of transfection, cells were stimulated with PMA/ionomycin for 6 h and luciferase activities were measured using the dual luciferase reporter assay system (Promega) according to the manufacturer's instructions. Firefly values were normalized to renilla values; Firefly values from luciferase reporter vectors, renilla values from pRL-TK.

**Chromatin accessibility assay.** Chromatin was isolated from in vitro-cultured T cells and digested with nuclease (Nse) mix using the EpiQuik chromatin accessibility assay kit (Epigentek). Isolated chromatin was divided into two, one for nuclease treatment, and another for non-treatment. After incubation at 37 °C for 4 min, DNA was purified followed by qPCR to amplify DNA fragment with primers for CNS-25 (same as ChIP primers) and $Il9$p or $Hbb\text{-}bs$ for negative control (F: 5′-gagtggca-cagcatccagggagaaa-3′, R: 5′-ccacaggccagagacagcagccttc-3′) The fold enrichment (FE) was calculated by the formula: $FE = 2\text{^}(Nse\ CT - no\ Nse\ CT) \times 100\%$.

**ChIP-seq analysis.** ChIP was performed as described above and high-throughput sequencing was performed by Array Star (Rockville, MD). Datasets are deposited in Gene Expression Omnibus (GEO; GSE106404). Raw ChIP-Sequence reads (replicates of p300 and Input (as control)) extracted from mouse Th9 cells in FASTQ formats. We examined the quality of these paired end ChIP-Seq datasets using the FASTX-Toolkit (http://hannonlab.cshl.edu/fastx_toolkit/index.html) and ensured the quality of sequence reads to a minimum quality score of 20 for each sample. We used HISAT (with default parameters)[64] for aligning the quality filtered sequence reads against mouse reference genome mm10. The aligned sequence data were collected in SAM (Sequence Alignment/Map) files obtained as outputs from HISAT. We employed MACS (version 1.4.2 201203052)[65], an efficient peak calling algorithm, with default parameters for the identification of p300 specific peaks genome-wide in aligned p300 and Input ChIP-seq samples (merged for corresponding replicates). Genomic distribution of p300-binding sites was analyzed by Pavis (https://manticore.niehs.nih.gov/pavis2/)[66] with the mm10 genomic annotation. In this annotation analysis we have considered 50 kb upstream and 10 kb downstream of peak sites to link all the p300-targeted genes where it could be potentially acting as $cis$ or distant regulatory element. Published ChIP-seq datasets (GSE22104 and GSE40463)[33,34] were analyzed and graphed in the same approach. DNA-binding motifs were identified using JASPAR and by hand with published consensus sequences. ChIP-seq datasets are deposited in GEO (GSE106404).

**Th9-enriched gene list.** RNA-seq results from Th9 cells (GSM1014577 and GSM978775)[22,67] were compared with RNA-seq data from other in vitro-derived Th subsets (2-week data from GSE48138)[68]. FASTQ files were aligned to reference file (mm9) and UCSC GTF file using tophat2 (tophat-2.0.13, parameter: no-novel-juncs [Quantification of reference annotation only]). FPKM values for each sample were calculated using Cufflinks (cufflinks-2.2.1, default). Cuffnorm (cufflinks-2.2.1, default settings) was used to combine and normalize FPKM values of each dataset. The unique high expression genes of each dataset were selected by looking at a 16× difference in expression and clustering analysis was performed on these selected lists using R's Pearson correlation.

**Statistical analysis.** Unless otherwise indicated, two-tailed Student's $t$ test or one-way analysis of variance was used to generate $p$-value data for all data. Post hoc Tukey test was used for multiple comparisons. $p \leq 0.05$ was considered statistically significant.

## Data availability

ChIP-seq datasets are deposited in GEO (GSE106404). All other data are available upon communication with the authors. A reporting summary for this Article is available as a Supplementary Information file.

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

## Acknowledgements

The authors thank Drs. Alexander Dent, Baohua Zhou, and Anthony Firulli for critical comments on this manuscript. We thank Drs. Peter Deak and Basar Bilgicer for providing the antibody used in the anaphylaxis studies. The work in the study was supported by NIH grants R01 AI057459, R01 AI129241, and R03 AI135356 to M.H.K. A.A.Q. was supported by T32 DK007519. Core facility usage was also supported by IU Simon Cancer Center Support Grant P30 CA082709 and U54 DK106846. Support provided by the Herman B Wells Center was in part from the Riley Children's Foundation.

## Author contributions

B.K., A.A.Q., B.J.U., and Y.F. performed experiments. R.S. and S.C.J. performed bioinformatics analysis. B.K., A.A.Q., and M.H.K. conceived and directed experiments, and wrote the paper.

## Additional information

**Competing interests:** The authors declare no competing interests.

