## [Peer Review file · Nature Communications]

Reviewers' comments:

Reviewer #1 (Remarks to the Author):

In the manuscript entitled „A conserved enhancer regulates IL9 expression in multiple lineages“ from Koh et al. the authors described an enhancer region upstream of the IL9 gene (CNS-25) that strongly promotes the expression of IL-9. Comparative analyses of different murine T cell subtypes revealed that CNS-25 preferentially influenced IL-9 transcription in Th9 and Tc9 cells. In addition, BMMCs and basophils exhibited a comparable regulation of IL-9 expression. With the aid of CNS-25-deficient mice the authors could confirm their findings by analyzing Th9-differentiation in vitro and by applying a Th9/mast cell-dependent disease model. Finally, it was shown that a human homologous sequence of CNS-25 termed CNS-18 showed identical activity in Th9 cells as described for its murine counterpart.

Comments:

This is a well written manuscript based on very detailed molecular analyses of the CNS-25 locus. In addition the generation of CNS-25-deficient mice and the comparative analyses of IL-9 expression using several CNS-25-deficient cell types as well as the usage of such mice in a preclinical model for allergy strongly suggest that CNS-25 is essential for high IL-9 expression. Finally, the analyses of the human homologous sequence CNS-18 revealed an identical function of this enhancer for the expression of human IL-9, at least in Th9 cells.

1. Concerning Fig. 2g the authors used the expression IL9 promoter which implicates that the human promoter was shown. Certainly, the murine one was shown and IL9 should be used.

2. Concerning the figure legends of Fig. 3 the authors stated that “the next closest gene to CNS-25, Fbxl21, was not detectably expressed in Th9 cells, and expression was not altered by CNS-25-deficiency (data not shown).”

If Fbxl21 was not detectably expressed in Th9 cells, how could they state that the expression was not altered by CNS-25-deficiency? Did they check this with a different cell type which expressed Fbxl21?

3. In the M&M section the authors described that they isolated human naïve CD4+ T cells by magnetic separation but they did not mention by which marker. If they used CD45RA they should be aware of the fact that terminally differentiated effector memory cells can re-express CD45RA (see: Larbi A1, Fulop T. From "truly naïve" to "exhausted senescent" T cells: when markers predict functionality. Cytometry A. 2014 Jan; 85(1): 25-35)

Reviewer #2 (Remarks to the Author):

Title: “A conserved enhancer regulates IL9 expression in multiple lineages“ by Byunghee Koh, Amina Abdul Qayum, Rajneesh Srivastava, Yongyao Fu, Benjamin J. Ulrich, Sarath Chandra Janga and Mark H. Kaplan.

In the manuscript by Byunghee Koh et al., the authors describe identification of a conserved regulatory element, termed CNS-25, that strongly enhances IL-9 production in mouse and human TH9 cells and other IL-9-producing cells. Here, the authors used state-of-the-art in vitro techniques as well as in vivo models to proof the relevance of CNS-25 in IL9/IL9 gene regulation.

In summary, this is a well-written manuscript demonstrating for the first time that CNS-25 represents an important regulatory element/enhancer of IL9/IL9 gene regulation in mice and humans. However, few concerns rose during the review:

Major critique

1. In Figure 4b the authors demonstrate that CNS-25 deficiency results in decreased binding of STAT5, STAT6, IRF4, BATF, and PU.1. In supplemental figure 2 b the authors demonstrate that CNS-25 deficiency does not affect expression of Batf, Irf4, Sfp1 (PU.1) and Stat6. Did the authors also analyze expression of Stat5? This is of particular importance, since IL-2 receptor- and STAT5-dependent signaling is essential for IL-9 production. In addition, IL-9 itself can transduce signals via IL-9 receptor and STAT5, suggesting an additional autocrine feedback loop on IL-9 production. Hence, the authors should be encouraged to analyze whether ectopic expression of STAT5 or enhanced STAT5 phosphorylation e.g. by addition of exogenous IL-2 (or IL-9) can compensate for decreased IL-9 production in absence of CNS-25.

2. Importantly, the authors identify a homologous CNS (CNS-18) as an enhancer of the human IL9 gene. Since in humans the IL9 gene is linked to the IL3, IL4, IL5 and IL13 locus, the authors should be encouraged to analyze the effect of CNS-18 deficiency on the regulation of these genes as well as on the effect of the respective cytokines (IL-3, IL-4, IL-5, IL-13) on IL-9 production in CNS-18-deficient and competent human T cells.

Minor critique

1. The manuscript should be checked for typos (e.g. "(Fig. 1a and Fig. S1b)") as well as for correct spelling of the mouse and human gene locus (e.g. Figure 2g: "IL9 promoter" would indicate the human but not the analyzed mouse (Il9) locus).

Reviewer #3 (Remarks to the Author):

In the paper entitled "a conserved enhancer regulates Il9 expression in multiple lineages", through p300 ChIP assay and bioinformatic analysis, the authors first identified CNS-25 as a potential enhancer for Il9 expression in T cells. Their further data showed that the CNS-25 site in Th9 cells were enriched with enhancer-specific histone markers, related key TFs and chromatin organizer proteins, and could drive Il9 promoter activity in IRF4-dependent manner. The authors then generated the CNS-25 KO mice, and found that CNS-25-deficiency not only greatly reduced IL-9 expression in a number of type 2 immune cells, such as Th9, Tc9, mast cells and basophils in vitro cultures, but also reduced airway hyperreactivity and IL-9 expression in allergic airway inflammation models. Moreover, the authors found that KO of the CNS-19 sequence, the mouse CNS-25 counterpart in human T cells, also decreased human Th9 differentiation, further consolidating the physiological relevance of their findings. The study is well performed and novel, and reveals an important cis-regulatory mechanism controlling IL-9 expression in multiple immune cell types.

Major concerns

- 1) Fig 2a, the authors showed a number of TFs interact with CNS-25. Do these TFs cluster together? the authors should further elucidate the biological relevance of this finding, such as by reporter gene assays with mutant CNS-25 constructs.
- 2) Fig 2f, the reporter assays were performed in EL4 cells, and the lineage specific activity of CNS25 need to be examined among different Th subsets.
- 3) Fig 4a/b, the effect of CNS-25-deficiency also affected the chromatin structure at CNS-6 and the Il9p regions, and binding of TFs, the authors need examine whether these sites were looping together in Th9 cells.

Minor concerns:

- 1) Fig 1c/2a-c/4a-b, the ChIP data all lacked IgG control. At least, the authors should provide the IgG control data in Th9 cells.
- 2) Fig 5a, individual ILC population with definitive markers should better be presented.
- 3) Fig 5f and 5g, do the authors know why the MC progenitors were differentiated affected by CNS9 in BM and spleens?

Reviewers' comments:

Reviewer #1 (Remarks to the Author):

In the manuscript entitled „A conserved enhancer regulates IL9 expression in multiple lineages“ from Koh et al. the authors described an enhancer region upstream of the IL9 gene (CNS-25) that strongly promotes the expression of IL-9. Comparative analyses of different murine T cell subtypes revealed that CNS-25 preferentially influenced IL-9 transcription in Th9 and Tc9 cells. In addition, BMMCs and basophils exhibited a comparable regulation of IL-9 expression. With the aid of CNS-25-deficient mice the authors could confirm their findings by analyzing Th9-differentiation in vitro and by applying a Th9/mast cell-dependent disease model. Finally, it was shown that a human homologous sequence of CNS-25 termed CNS-18 showed identical activity in Th9 cells as described for its murine counterpart.

Comments:

This is a well written manuscript based on very detailed molecular analyses of the CNS-25 locus. In addition the generation of CNS-25-deficient mice and the comparative analyses of IL-9 expression using several CNS-25-deficient cell types as well as the usage of such mice in a preclinical model for allergy strongly suggest that CNS-25 is essential for high IL-9 expression. Finally, the analyses of the human homologous sequence CNS-18 revealed an identical function of this enhancer for the expression of human IL-9, at least in Th9 cells.

1. Concerning Fig. 2g the authors used the expression IL9 promoter which implicates that the human promoter was shown. Certainly, the murine one was shown and IL9 should be used.

We thank the reviewer for pointing this out and we have fixed the notations.

2. Concerning the figure legends of Fig. 3 the authors stated that “the next closest gene to CNS-25, Fbxl21, was not detectably expressed in Th9 cells, and expression was not altered by CNS-25-deficiency (data not shown).“

If Fbxl21 was not detectably expressed in Th9 cells, how could they state that the expression was not altered by CNS-25-deficiency? Did they check this with a different cell type which expressed Fbxl21?

We apologize for the lack of clarity in the previous text. To further address this point, we measured Fbxl21 expression in all T cell subsets, mast cells, bone marrow and we did not observe expression. In brain tissue, we did observe Fbxl21 expression, but it was not altered by CNS-25 deficiency. These observations are noted more clearly in the text.

3. In the M&M section the authors described that they isolated human naïve CD4+ T cells by magnetic separation but they did not mention by which marker. If they used CD45RA they should be aware of the fact that terminally differentiated effector memory cells can re-express CD45RA (see: Larbi A1, Fulop T. From "truly naïve" to "exhausted senescent" T cells: when markers predict functionality. Cytometry A. 2014 Jan;85(1):25-

35)

First, we have now noted in the methods that we did select on CD45RA and we thank the reviewer for making this point. Second, we have now examined naïve T cell purity in our studies and it was over 95% CD4⁺ CD45RA⁺. Of the CD4⁺ CD45RA population, we further examined expression of KLRG1 and CD57, the markers identified in the report noted by the reviewer as indicators of the ‘exhausted senescent’ cells. We observed less than 1% of cells that expressed either marker. Thus, while this could have an impact, we think this is likely minor. Still, we have included data to make this point in Supplemental Figure 5b, and have noted this information in the text.

Reviewer #2 (Remarks to the Author):

Title: “A conserved enhancer regulates Il9 expression in multiple lineages“ by Byunghee Koh, Amina Abdul Qayum, Rajneesh Srivastava, Yongyao Fu, Benjamin J. Ulrich, Sarath Chandra Janga and Mark H. Kaplan.

In the manuscript by Byunghee Koh et al., the authors describe identification of a conserved regulatory element, termed CNS-25, that strongly enhances IL-9 production in mouse and human TH9 cells and other IL-9-producing cells. Here, the authors used state-of-the-art in vitro techniques as well as in vivo models to proof the relevance of CNS-25 in Il9/IL9 gene regulation.

In summary, this is a well-written manuscript demonstrating for the first time that CNS-25 represents an important regulatory element/enhancer of Il9/IL9 gene regulation in mice and humans. However, few concerns rose during the review:

Major critique

1. In Figure 4b the authors demonstrate that CNS-25 deficiency results in decreased binding of STAT5, STAT6, IRF4, BATF, and PU.1. In supplemental figure 2 b the authors demonstrate that CNS-25 deficiency does not affect expression of Batf, Irf4, Sfp1 (PU.1) and Stat6. Did the authors also analyze expression of Stat5? This is of particular importance, since IL-2 receptor- and STAT5-dependent signaling is essential for IL-9 production. In addition, IL-9 itself can transduce signals via IL-9 receptor and STAT5, suggesting an additional autocrine feedback loop on IL-9 production. Hence, the authors should be encouraged to analyze whether ectopic expression of STAT5 or enhanced STAT5 phosphorylation e.g. by addition of exogenous IL-2 (or IL-9) can compensate for decreased IL-9 production in absence of CNS-25.

The reviewer makes several important points and we have performed the following experiments to address them.

1. We have added expression analysis of Stat5a and Stat5b to Supplementary Fig. 3b. There is no difference in expression in the CNS-25 mutant mice.
2. While IL-9 can activate STAT5 in some settings, it does not have an appreciable feedback effect on Th9 differentiation. IL-9 can be added or blocked during the Th9 differentiation process without an effect on Th9 differentiation, and in fact *Il9r* expression on T cells, at least in vitro, is relatively low. Moreover, since IL-2, a far more

potent STAT5 activator than IL-9, is added to the cultures, we would not expect STAT5 activation to be deficient. We've added a comment on this in the discussion.

3. We have performed transduction with the constitutively active Stat5 retrovirus, and this data is now in Fig. 4g. It shows that active Stat5 can increase IL-9 production, even in the absence of CNS-25. However, even with the active Stat5, CNS-25 mutant Th9 cells make less IL-9 than transduced wild type cells. Thus, even supraphysiological levels of active STAT5 do not overcome the contribution of CNS-25.

2. Importantly, the authors identify a homologous CNS (CNS-18) as an enhancer of the human IL9 gene. Since in humans the IL9 gene is linked to the IL3, IL4, IL5 and IL13 locus, the authors should be encouraged to analyze the effect of CNS-18 deficiency on the regulation of these genes as well as on the effect of the respective cytokines (IL-3, IL-4, IL-5, IL-13) on IL-9 production in CNS-18-deficient and competent human T cells.

We analyzed the expression of IL3, IL4, IL5 and IL13 in sorted Th9 cells. IL4 expression was not detected in control or CNS-18-deficient culture. In our analysis of the expression of the other cytokines (now included as Supplementary Fig. 5a), we observed no significant differences in expression of those cytokines between control and CNS-18 deleted cells. As such, we do not think there are any indirect effects of CNS-18-deficiency on IL9 production, and this is noted in the text.

Minor critique

1. The manuscript should be checked for typos (e.g. "(Fig. 1a and Fig. S1b)) as well as for correct spelling of the mouse and human gene locus (e.g. Figure 2g: "IL9 promoter" would indicate the human but not the analyzed mouse (Il9) locus.

We have extensively edited the manuscript.

Reviewer #3 (Remarks to the Author):

In the paper entitled "a conserved enhancer regulates Il9 expression in multiple lineages", through p300 ChIP assay and bioinformatic analysis, the authors first identified CNS-25 as a potential enhancer for Il9 expression in T cells. Their further data showed that the CNS-25 site in Th9 cells were enriched with enhancer-specific histone markers, related key TFs and chromatin organizer proteins, and could drive Il9 promoter activity in IRF4-dependent manner. The authors then generated the CNS-25 KO mice, and found that CNS-25-deficiency not only greatly reduced IL-9 expression in a number of type 2 immune cells, such as Th9, Tc9, mast cells and basophils in vitro cultures, but also reduced airway hyperreactivity and IL-9 expression in allergic airway inflammation models. Moreover, the authors found that KO of the CNS-19 sequence, the mouse CNS-25 counterpart in human T cells, also decreased human Th9 differentiation, further consolidating the physiological relevance of their findings. The study is well performed and novel, and reveals an important cis-regulatory mechanism controlling IL-9 expression in multiple immune cell types.

Major concerns

1) Fig 2a, the authors showed a number of TFs interact with CNS-25. Do these TFs cluster together? the authors should further elucidate the biological relevance of this finding, such as by reporter gene assays with mutant CNS-25 constructs.

We are not exactly clear what the reviewer is referring to by 'cluster together'. Even in the 286 bp *IL9* CNS-25.1 segment, the binding sites for the CNS-25-binding factors are spread throughout the sequence, despite the co-localization by ChIP within that region. We do know from our publications, and many other publications in the field, that deficiency in the factors that bind CNS-25 (BATF, IRF4, STAT5, STAT6, GATA3, Foxo1) severely affects IL-9 production. We think that more convincing than using reporter assays to address this point is to examine the endogenous locus, and we have tested the co-dependency of factors binding to the enhancer using gene-deficient cells and ChIP assay. We observed that in IRF4-deficient and STAT6-deficient Th9 cells there is significantly decreased binding of BATF and p300 to CNS-25, and also IRF4 in the STAT6-deficient cells. The activating chromatin modifications are also diminished in IRF4- or STAT6-deficient Th9 cells. This suggests that there is a co-dependency in the interaction of factors at CNS-25 and we have added a point on this in the discussion. We agree with the reviewer that this is an important point, and we think it requires more extensive experimentation that will extend beyond this initial report.

2) Fig 2f, the reporter assays were performed in EL4 cells, and the lineage specific activity of CNS25 need to be examined among different Th subsets.

We performed reporter assays using in vitro cultured Th0 and Th9 cells. In both Th0 and Th9 cells, CNS-25 significantly enhance *IL9* promoter-directed reporter activity. This new data, and the previous reporter assay data are now in Supplemental Fig. 2.

3) Fig 4a/b, the effect of CNS-25-deficiency also affected the chromatin structure at CNS-6 and the *IL9*p regions, and binding of TFs, the authors need examine whether these sites were looping together in Th9 cells.

We performed a chromosome conformation capture assay (3C assay) to demonstrate the physical interaction between CNS-25 and the *IL9* promoter in Th2 and Th9 cells. Using *IL9* promoter as an anchor, we observed a significantly higher interaction frequency at the CNS-25 in Th9 cells than in Th2 cells, with no difference in the interaction frequency (and lower interaction frequency overall) in the adjacent DNA regions at -26 and -23 kb. This new data is included as Fig. 2d.

Minor concerns:

1) Fig 1c/2a-c/4a-b, the ChIP data all lacked IgG control. At least, the authors should provide the IgG control data in Th9 cells.

IgG controls were of course performed for all experiments. We have presented the data with IgG control values subtracted from the specific ChIP values, as we have done in many other previous reports. To make this point clearer, we have included a graph in Fig. 1d to show an example of control Ig values for the p300 ChIP, and have noted that background values are subtracted in subsequent analyses.

2) Fig 5a, individual ILC population with definitive markers should better be presented.

We have added “Fig S4a” to show the gating strategy for analyzing ILC2 populations in IL-33 treated mice.

3) Fig 5f and 5g, do the authors know why the MC progenitors were differentiated affected by CNS9 in BM and spleens?

While we have not entirely defined why there is a different response in these two populations, they are entirely different populations as described in the Methods, Innate cell isolation section. Bone marrow mast cell progenitors (MCp) were gated as lineage negative (CD5, B220, CD11b, CD27, anti-Gr-1(Ly6G/C), Ly6C, Sca-1, Ter119, CD19, NK1.1), FcεR1^{low}c-Kit⁺ST2⁺β7⁺. Basophil/mast cell progenitors (BMCp) in the spleen were gated as lineage negative (B220, CD3, Ly6C/G, NK1.1, GR1, Ter119, CD5) c-Kit⁺FcγRII/III^{hi}β7^{hi}ST2⁺. These designations are based on published work as cited in the report. There is additional literature showing that in some models one population is altered without affecting the other (Toyoshima et al, Int. Immunol. 2017). A truly definitive understanding of this observation requires a further understanding of mast cell development. This is certainly of interest, but not something pursued in this report.

REVIEWERS' COMMENTS:

Reviewer #1 (Remarks to the Author):

The authors have satisfactorily addressed the comments of the review; therefore, publication of the revised version of this manuscript can be recommended.

Reviewer #2 (Remarks to the Author):

In the revised manuscript by Byunghee Koh et al., "A conserved enhancer regulates II9 expression in multiple lineages" the authors have responded to the criticisms raised with inclusion of additional data further substantiating the important role of the identified conserved regulatory element (CNS-25) for II9 locus regulation.

Overall the manuscript is greatly improved and I would like to recommend it for publication in Nature Communications.

Reviewer #3 (Remarks to the Author):

The authors have sufficiently addressed my previous critiques.

Response to Reviewers

As noted below, the three reviewers had no further comments to address in revision.

Reviewer #1 (Remarks to the Author):

The authors have satisfactorily addressed the comments of the review; therefore, publication of the revised version of this manuscript can be recommended.

Reviewer #2 (Remarks to the Author):

In the revised manuscript by Byunghye Koh et al., "A conserved enhancer regulates I19 expression in multiple lineages" the authors have responded to the criticisms raised with inclusion of additional data further substantiating the important role of the identified conserved regulatory element (CNS-25) for I19 locus regulation.

Overall the manuscript is greatly improved and I would like to recommend it for publication in Nature Communications.

Reviewer #3 (Remarks to the Author):

The authors have sufficiently addressed my previous critiques.